# A Controlled Study on Long Context Extension and Generalization in LLMs

Yi Lu[2*]   Jing Nathan Yan[1*]   Songlin Yang[3]   Justin T. Chiu[1]
Siyu Ren[2]   Fei Yuan[2]   Wenting Zhao[1]   Zhiyong Wu[2+]   Alexander M. Rush[1]
[1]Cornell University, [2]Shanghai AI Lab, [3]Massachusetts Institute of Technology

## Abstract

Achieving robust textual comprehension and in-context learning requires language models capable of interpreting entire document contexts. However, scaling these models directly to long contexts remains technically challenging, prompting a surge of "extension" strategies. To date, rigorous comparisons among these approaches have been complicated by inconsistent base models, training data, and evaluation metrics, limiting our understanding of how long-context performance may differ from standard benchmarks. In this work, we introduce a *controlled extension protocol* and a *standardized evaluation* pipeline, enabling an apples-to-apples comparison across diverse long-context methods. Through extensive experiments, we uncover three key insights: (1) perplexity emerges as a helpful (albeit imperfect) indicator for gauging model quality on lengthy-context tasks, (2) approximate attention mechanisms exhibit systematic performance deficits on long-context benchmarks, and (3) exact fine-tuning remains robust within its extension range, although extrapolation beyond that range continues to pose challenges. Our results not only help clarify the current landscape of long-context modeling but also offer guidance for building more capable, context-aware language models. To fostering transparency and accelerating progress in this critical area of AI research, all codebases, trained models, and checkpoints are made available open-source via https://github.com/Leooyii/LCEG.

## 1  Introduction

The scale of pretraining data for large language models (LLMs) has grown dramatically, with open-source models now trained on up to 15 trillion tokens (AI@Meta, 2024). Despite this progress, *implementation challenges* often hinder fully training models with larger context windows (Liu et al., 2023a). Yet *long-context capabilities* are increasingly recognized as essential for tasks demanding extensive textual understanding, such as referencing entire textbooks (Tanzer et al., 2024), summarizing novels (Kryściński et al., 2022), or performing many-shot learning (Bertsch et al., 2024; Li et al., 2023b).

To circumvent the difficulty of long-context pretraining, researchers have proposed *context extension* methods. These approaches adapt LLMs pretrained on standard sequence lengths to much larger context windows (Chen et al., 2023a; Peng et al., 2023; Han et al., 2023; bloc97, 2023), differing in attention mechanisms and adaptation procedures. However, significant variability in training complexity, data usage, and model performance often leads to *inconsistent* or *incomplete* experimental comparisons.

Existing studies have introduced specialized metrics—such as long-context perplexity (Chen et al., 2023a;b; Han et al., 2023; Hsieh et al., 2024) and retrieval accuracy (Mohtashami & Jaggi, 2023; gkamradt, 2023)—to capture extended-context performance (Bai et al., 2023; An et al., 2023). Yet their utility is difficult to calibrate across diverse extension methods. Further complicating matters, prior work typically focuses on distinct base models, different post-training data, or custom evaluation protocols, making it challenging to *directly* compare reported results.

---

*Equal contribution. [+] Correspondence author

In this work, we implement a controlled protocol for context extension. The aim is to compare context extension while removing spurious factors that impact LLM ability.

**Unified Mathematical Framework.** We consolidate multiple mathematical formulations of long-context adaptation into a single, well-rounded perspective, addressing fragmentation in existing literature and offering a rigorous theoretical grounding for context extension.

**Controlled Protocol.** We propose a carefully designed experimental setup that employs five *identical* open-weight base models, same training data, and tuning hyperparameters across a diverse set of extension methods. This removes confounding factors and permits an apples-to-apples comparison of each approach's ability to scale to longer contexts.

**Robust Evaluation.** Our study incorporates both intrinsic (e.g., perplexity) and extrinsic (e.g., downstream tasks) metrics, measured *within* and *beyond* each method's intended extension range. Although perplexity only partly reflects real-world performance, it still provides useful insights for many tasks, albeit less reliably when approximate attention methods are involved.

**Key Findings.** First, while some question perplexity's suitability for long-context evaluation, we find it often correlates with downstream performance in controlled settings, though its predictive power is not universal. Second, approximate attention approaches show systematic performance deficits, raising questions about their trade-offs between efficiency and accuracy. Third, continual fine-tuning with exact attention reliably enhances long-context performance, especially within the intended extension range. By contrast, *extrapolation* to even longer windows remains an open challenge, suggesting opportunities for future work.

Taken together, our unified mathematical framework, standardized experiments, and multi-metric evaluation illuminate the landscape of long-context modeling. By bridging existing gaps in methodology and metrics, we hope to provide a common reference point and spark more consistent research into extended context windows in LLMs.

## 2 Related Work

**Long Context Methods** We divide extension methods into three broad classes: exact attention, approximate attention, and context compression. Exact attention methods augment the parameterization of attention. Position interpolation (PI) (Chen et al., 2023a), NTK-aware (bloc97, 2023), Dynamic NTK (emozilla, 2023), YaRN (Peng et al., 2023), and CLEX (Chen et al., 2024), all based on RoPE (Su et al., 2021), design position embeddings for length extension. These methods may be applied with fine-tuning or to frozen models. Other exact attention methods focus on training-time improvements, such as contrastive training (Tworkowski et al., 2023). Approximate attention methods uses structured attention approximations to minimize the computational cost of length growth. Chen et al. (2023b) uses LoRA (Hu et al., 2021) and a specialized local attention mechanism to reduce further the computational overhead of further fine-tuning with long context. Other approaches break the text into chunks and utilize a well-designed "chunk representation" to retrieve relevant chunks for attention (Mohtashami & Jaggi, 2023; Xiao et al., 2024; Lu et al., 2024). LM-Infinite and StreamLLM (Han et al., 2023; Xiao et al., 2023) retain only a few tokens from the beginning of the text and a local window to keep the attention window within the pretrained length. Xu et al. (2024) focuses on using retrievers to retrieve relevant blocks from long documents. Finally, context compression methods, which we do not explore in this work, reduce length extension to length compression via a summarization step (Jiang et al., 2023; Li et al., 2023c).

**Long Context Evaluation Benchmarks** The Long Range Arena (LRA) (Tay et al., 2020) is an early efforts evaluating the proficiency of processing long contexts. Since then, a growing number of benchmarks have emerged, including LongBench (Bai et al., 2023), LEval (An et al., 2023), and LooGLE (Li et al., 2023a). These benchmarks are a mixture of diverse downstream tasks explicitly tailored to assess the capabilities of LLMs in understanding and generating lengthy contexts. Among these benchmarks, LongBench stands out for its inclusion of diverse sequences with varying lengths, distributions, patterns, languages, and domains, enabling a comprehensive, nuanced evaluation. In addition to evaluating LLMs' performance on downstream NLP tasks, there is another line of benchmarks that specifically focuses on assessing particular aspects of long context processing ability Liu et al. (2023b); Hsieh et al. (2024). For instance, Mohtashami & Jaggi (2023) propose the

passkey retrieval task to challenge a language model to accurately locate and retrieve a simple passkey (a five-digit random number) in a long context sequence. Similarly, the Needle in a Haystack (gkamradt, 2023) test requires the model to accurately recite the information from a specified sentence(the "needle"). However, most existing works mainly focus on evaluating mainstream commercial models (e.g. GPT-4 and Claude), open-source base models, or just perform individual evaluations of a few long context methods. There is a lack of comprehensive, yet controlled evaluation on long-context extension techniques.

## 3    Context Extension Methods

### 3.1    Background: Attention and RoPE

The bottleneck in long context modeling in Transformers is attention. Attention is defined over $C$ embeddings $\mathbf{X} = [\mathbf{x}_1, \mathbf{x}_2, \dots, \mathbf{x}_C]^\top \in \mathbb{R}^{C \times d}$ where $d$ is the model dimension. Learned weight matrices $\mathbf{W}_v \in \mathbb{R}^{d \times d_k}$, $\mathbf{W}_q \in \mathbb{R}^{d \times d_k}$, and $\mathbf{W}_k \in \mathbb{R}^{d \times d_k}$ are used to transform these inputs where $d_k$ is the projected hidden dimension. The attention mechanism itself computes the attention matrix and applies it to produce a weighted sum of the value vectors:

$$\text{Attention}(\mathbf{Q}, \mathbf{K}, \mathbf{V}) = \mathbf{A}\mathbf{V} = \text{softmax}\left(\frac{\mathbf{Q}\mathbf{K}^\top}{\sqrt{d_k}}\right)\mathbf{V}. \tag{1}$$

Basic attention was originally defined with: $\mathbf{Q} = \mathbf{X}\mathbf{W}_q, \mathbf{K} = \mathbf{X}\mathbf{W}_k, \mathbf{V} = \mathbf{X}\mathbf{W}_v$. However, this approach does not directly encode the relative position of keys and values.

Rotary Position Embeddings (RoPE) (Su et al., 2024) encode positional information by applying a phase rotation to each element of the embedding vectors. Formally, we define a transformation $\mathbf{f}$:

$$\mathbf{f}_\mathbf{W}(\mathbf{x}_i, \boldsymbol{\theta}) = \mathbf{R}(\boldsymbol{\theta}, i)\mathbf{W}^\top \mathbf{x}_i \tag{2}$$

Here $\mathbf{x}_i \in \mathbb{R}^{d_k}$ is an embedding for position $i$, $\mathbf{W}$ is a projection matrix, and $\boldsymbol{\theta} \in \mathbb{R}^{d_k/2}$ is a frequency basis. The function is defined based on the rotary position matrix:

$$\mathbf{R}(\boldsymbol{\theta}, i) = \begin{pmatrix} \cos i\theta_1 & -\sin i\theta_1 & \cdots & 0 & 0 \\ \sin i\theta_1 & \cos i\theta_1 & \cdots & 0 & 0 \\ \vdots & & & & \\ 0 & 0 & \cdots & \cos i\theta_{\frac{d_k}{2}} & -\sin i\theta_{\frac{d_k}{2}} \\ 0 & 0 & \cdots & \sin i\theta_{\frac{d_k}{2}} & \cos i\theta_{\frac{d_k}{2}} \end{pmatrix} \tag{3}$$

Due to the arrangement of frequencies, this matrix has the property that $\mathbf{R}(\boldsymbol{\theta}, n - m) = \mathbf{R}(\boldsymbol{\theta}, m)^\top \mathbf{R}(\boldsymbol{\theta}, n)$ by Ptolemy's identity. We redefine the query-key product between two positions $m$ and $n$ as,

$$\mathbf{q}_m^\top \mathbf{k}_n = \mathbf{f}_{\mathbf{W}_q}(\mathbf{x}_m, \boldsymbol{\theta})^\top \mathbf{f}_{\mathbf{W}_k}(\mathbf{x}_n, \boldsymbol{\theta}) \tag{4}$$

$$= \left(\mathbf{R}(\boldsymbol{\theta}, m)\mathbf{W}_q^\top \mathbf{x}_m\right)^\top \left(\mathbf{R}(\boldsymbol{\theta}, n)\mathbf{W}_k^\top \mathbf{x}_n\right) \tag{5}$$

$$= \mathbf{x}_m^\top \mathbf{W}_q \mathbf{R}(\boldsymbol{\theta}, n - m)\mathbf{W}_k^\top \mathbf{x}_n \tag{6}$$

In this way, the relative positional information $n - m$ is implicitly injected into the query and key product, thus the attention score. The standard RoPE transformation, $f_\mathbf{W}(x_i, \boldsymbol{\theta})$, sets the components $\theta_j = b^{-\frac{2j}{d_k}}$ with base $b = 10000$.

### 3.2    Adjusting the Frequency of RoPE for Long Context Extension

We consider four methods for performing length extension on RoPE embeddings: Position Interpolation (PI) (Chen et al., 2023a), NTK-RoPE (emozilla, 2023), YaRN (Peng et al., 2023) and CLEX (Chen et al., 2024). In this section our goal is to extend a method trained to extend position embeddings for context length $C$ to length $C' >> C$. The methods in this section perform this extension by scaling the frequencies with the *base scaling vector* $\boldsymbol{\alpha} \in \mathbb{R}^{\frac{d_k}{2}}$:

$$\mathbf{f}_\mathbf{W}(x_i) = \mathbf{f}(x_i, \boldsymbol{\alpha} \odot \boldsymbol{\theta}). \tag{7}$$

**Linear Position Interpolation (PI)** decreases the frequencies of the basis functions so that more tokens fit within each period. PI set the components of the base scaling vector to

$$\alpha_j^{\text{PI}} = \frac{C}{C'} = \frac{1}{t}. \tag{8}$$

where $t = \frac{C'}{C}$ is target length ratio. PI has been integrated into many open-source models such as LLaMA2-7B-32K (Together.AI, 2023), Vicuna-7B-v1.5 (Chiang et al., 2023).

**Neural Tangent Kernel Interpolation RoPE (NTK-RoPE)** builds on linear position interpolation by introducing a per-dimension scaling factor. Inspired by findings from the NTK literature that show that high-frequency features are difficult for MLPs to learn, NTK-RoPE preserves high-frequency features while extending the period of low-frequency features. This is accomplished via a dimension-dependent base scaling vector $\boldsymbol{\alpha}$:

$$\alpha_j^{\text{NTK-RoPE}} = \kappa^{-\frac{2j}{d_k}}, \tag{9}$$

where $\kappa = (t)^{\frac{d_k}{d_k-2}}$ so that the lowest frequency is scaled to match PI and the highest frequency remains the same as in RoPE. An extension to this approach, Dynamic NTK-RoPE suggests that instead of fixing scaling based on a set ratio $s$ for all examples during inference, the formula should adapt to the current context length for a specific example. We followed the set up of Fu et al. (2024) for Dynamic NTK-RoPE. More details are in the Appendix H.

**YaRN**, another RoPE extension method, uses "NTK-by-parts" interpolation strategies across different dimensions of the embedding space and introduces a temperature factor to adjust the attention distribution for long inputs.

$$\alpha_j^{\text{YaRN}} = \left( (1 - \gamma_j) \frac{1}{t} + \gamma_j \right) / \sqrt{T} \tag{10}$$

We use a ramp vector $\gamma$ to determine the interpolation between the $\frac{1}{t}$ and the original frequency base. The interpolation gating is set based on the frequency for the dimension $j$. More details about this ramp function can be found in the Appendix F.

Other methods such as **CLEX** Chen et al. (2024) models the scaling vectors as a dynamical system, with the goal of learning target-length dependent scaling vectors.

### 3.3 Adjusting Attention for Context Extension

An alternative approach is to modify the attention function itself. Approaches to handling longer contexts fall into two main categories: approximate attention and attention modification. In approximate attention, instead of computing the full attention matrix, methods select a subset of positions to attend to. In attention modification, the approach incorporates additional information through retrieval or other mechanisms. We examine three methods across these categories: sparse attention, sliding window attention, and retrieval attention.

**LongLoRA** (Chen et al., 2023b) avoids computing attention ranges over $C'$ by only computing the block-diagonl part of attention. Formally, given a sequence length of $C'$, LongLoRA divides it into $M$ blocks of size $B$, resulting in a sparse attention matrix $\mathbf{A} \in \mathbb{R}^{C' \times C'}$ with a block-diagonal structure:

$$\mathbf{A} = \begin{bmatrix} \mathbf{A}_1 & \mathbf{0} & \cdots & \mathbf{0} \\ \mathbf{0} & \mathbf{A}_2 & \cdots & \mathbf{0} \\ \vdots & \vdots & \ddots & \vdots \\ \mathbf{0} & \mathbf{0} & \cdots & \mathbf{A}_M \end{bmatrix} \tag{11}$$

where $\mathbf{A}_i \in \mathbb{R}^{B \times B}$ is the attention matrix for the $i$-th block. In addition, they shift the blocks for half of the heads enabling the information flow between groups via shifting. Notably, while they employ local attention during the fine-tuning phase, full attention is still adopted during the inference stage.

**Landmark Attention** (Mohtashami & Jaggi, 2023) addresses the challenge of attending over long sequences by breaking the input sequence into chunks and using trainable "landmark" tokens to summarize these chunks. The attention process is carried out in two stages. Given

a sequence of $C'$ embeddings, divided into $M$ chunks, each of length $B$, the first step is to compute global attention between the query vectors $\mathbf{Q} \in \mathbb{R}^{C' \times d_k}$ (corresponding to all input embedding) and the landmark vectors $\mathbf{L} \in \mathbb{R}^{M \times d_k}$ (which represent the chunks). From this global attention, a set of $n$-most attended-to chunks is selected for further processing. Next, a local attention mechanism is applied within each of the selected chunks. For the $n$-th selected chunk, the key matrix for the chunk is denoted as $\mathbf{K}_n \in \mathbb{R}^{B \times d_k}$ and $\mathbf{Q}_n \in \mathbb{R}^{B \times d_k}$. The attention matrices are then computed as follows:

$$\mathbf{A}^1 = \text{softmax}\left(\frac{\mathbf{Q}\mathbf{L}^T}{\sqrt{d_k}}\right) \in \mathbb{R}^{C' \times M}, \mathbf{A}^{2,n} = \text{softmax}\left(\frac{\mathbf{Q}_n\mathbf{K}_n^T}{\sqrt{d_k}}\right) \in \mathbb{R}^{B \times B}, \tag{12}$$

The final attention for each embedding is a combination of these two attentions, which efficiently scales attention mechanisms for long sequences by focusing on landmark tokens that summarize large parts of the sequence, followed by local attention within the relevant chunks.

**LM-Infinite** (Han et al., 2023) (a.k.a., Sliding Window Attention) maintains a sliding local window of size $M$ along with a fixed global memory of $G$ positions at the starting point of the given embedding. Given $C'$ embeddings, attention is computed over the $M$ embeddings in its local window and $G$ embeddings in global memory. LM-Infinite replaces relative positional information $n - m$ with $\min(n - m, C)$ while computing the query and key product in Eq 4. Altogether, LM-Infinite reduces the complexity from $\mathcal{O}((C')^2)$ to $\mathcal{O}(C'(M + G))$ without the need to scale positional encoding.

**Self-Extend** (Jin et al., 2024) maps the unseen positions in extended context length $C'$ to positions within the pretraining context length $C$ to avoid training. For each embeddings, Self-Extend chooses closest $M$ embeddings and any embeddings beyond are divided into multiple groups. Each group contains $N$ embeddings. When performing query-key product between two positions $m$ and $n$ in Equation 4, the relative positional information $n - m$ is replaced by $r$ which is computed by scaling $n - m$ w.r.t $M$ and $N$:

$$r = \begin{cases} n - m, & n - m \leq M, \\ M + \lfloor \frac{n-m}{N} \rfloor - \lfloor \frac{M}{N} \rfloor, & n - m > M. \end{cases} \tag{13}$$

where $\lfloor \cdot \rfloor$ denotes the floor division. Extended context length $C'$ is $(C - M) \cdot N + M$.

## 4 Long-Context Extension Protocol

**Base Model** All models start from an identical base checkpoint. We choose to use five different base models LLaMA2-7B, 13B, 70B (Touvron et al., 2023), Phi-2(Javaheripi et al., 2023), LLaMA3-8B(Dubey et al., 2024)for context extension experiments, to verify whether the trends and analyses we observed are consistent across different base models, thereby avoiding potential over-generalization. **Note that in our main findings, we only report results with LLaMA2-7B base model** to maintain conciseness and avoid redundancy as we find most of general findings from LLaMA2-7B can be transferred to all other models. Results from other models are provided in Appendix G.

**Fine-Tuning** We sample 1B tokens from a long-context data mixture following Fu et al. (2024). The data details are reported in Appendix I. We focus on extending the context window from 4k to 32k since most benchmarks require contexts under 32k. We maintain a fixed training recipe to ensure consistency across all models (Chen et al., 2023b). We follow existing practices by keeping an exponential moving average (EMA) of model weights with a constant decay and a linear learning rate warm-up. Most training hyperparameters are based on (Fu et al., 2024), with the learning rate set to $2 \times 10^{-5}$. Our experiments are done on 8 NVIDIA A100 GPUs. Detailed hyperparameter setups are in Appendix H.

**Metrics** We consider two sets of intrinsic metrics. The first is based on *perplexity*. We use the book corpus PG19 (Rae et al., 2019) and the Proof-pile dataset (Azerbayev et al., 2023) to evaluate the long sequence language modeling performances. Following Press et al. (2022), all perplexity evaluations are calculated using a sliding window with a window size of 256.

The second is based on *retrieval*. We focus on the needle in the haystack task (gkamradt, 2023)(NIAH). NIAH involves identifying a specific, relevant piece of information (the

Table 1: Overview of results across different extension types.

| Attention Mechanisms | | Model | PPL | ND | Mshots | LB | RULER | HEL |
|---|---|---|---|---|---|---|---|---|
| Exact Attention | Frozen | NTK-F | 14.5 | 18.8 | 64.5 | 25.5 | 0.7 | 3.5 |
| | Fine-Tuned | PI | 5.9 | 42.1 | **75.5** | 33.5 | 57.7 | 25.7 |
| | | YaRN | 5.9 | 46.7 | 75.0 | 33.5 | 37.0 | 21.3 |
| | | CLEX | 5.8 | 71.1 | 74.0 | 33.5 | 52.2 | 26.4 |
| | | NTK-32K | **5.8** | **83.7** | 71.0 | **35.3** | 59.4 | 28.3 |
| | | NTK-64K | 5.9 | 69.1 | 73.0 | 34.3 | **60.0** | 29.3 |
| Modified. Attention | Modified. Attention | LM-Infinite | 6.7 | 23.9 | 61.5 | 25.8 | 12.3 | - |
| | | Self-Extend | 6.1 | 25.8 | 72.0 | 33.6 | 29.5 | 19.7 |
| | Approxi. Attention | LongLora | 9.9 | 20.3 | 55.5 | 23.3 | 3.5 | - |
| | | Landmark | 8.1 | 50.9 | 50.0 | 28.2 | 13.6 | - |

"needle") within a large set of irrelevant data (the "haystack"). This task is commonly used to test the precision and recall capabilities of LLMs in scenarios where the relevant data is sparse and surrounded by a significant amount of noise. We also evaluate with RULER (Hsieh et al., 2024). RULER enhances the standard NIAH test by incorporating variations with different types and quantities of needles with new task categories, such as multi-hop tracing and aggregation.

For extrinsic metrics, we consider a collection of tasks. LongBench (Bai et al., 2023) is a family of bilingual, multitask evaluations for long-context understanding widely used in measuring the long-context abilities of LLMs (Jin et al., 2024; Xiao et al., 2024; Lu et al., 2024). LongBench includes single-document question answering, multi-document QA, summarization, few-shot learning, and code completion. We follow Bai et al. (2023) to evaluate the models on 32k context window sizes by truncating the prompt from the middle when the task length exceeds a designated context window size. We also consider the ManyShots tasks, where the long-context model will be given several examples as prompts. We use the Trec News (Li & Roth, 2002) dataset for this task. Additionally, we evaluate HELMET (Yen et al., 2025), a comprehensive benchmark encompassing seven diverse, application-centric categories with controllable lengths up to 128k tokens.

## 5 Experimental Results and Analysis

### 5.1 Result Overview

Table 1 overviews the results across both types of evaluation. The main result demonstrate that fine-tuned exact attention methods for long contexts, such as NTK-32K and YARN, consistently outperform approximate attention methods by a significant margin. This suggests that trading accuracy for speed in approximate attention methods can result in a loss of important reasoning capabilities, particularly for retrieval-based tasks. The performance disparity highlights the importance of exact attention in maintaining high accuracy over extended contexts, emphasizing the need for careful consideration of attention type in model design for long-context tasks. We now consider each type of result in detail.

### 5.2 Intrinsic tasks

**Perplexity** Table 2 shows perplexity scores across length. We see that continuous fine-tuning methods like PI, YaRN, and LongLora effectively keep low perplexity scores within the pre-training context length. However, when the context length exceeds perplexity scores escalate once the context surpasses the pre-trained window. Only NTK and CLEX can generalize to unseen sequence length in both pretraining and continual finetuning. Additionally, we find that exact attention maintains better perplexity than LoRA, which may reduce LongLora's ability. We also note that results on both PG19 and Proof-file gave nearly consistent conclusions.

**Needle-in-the-haystack** NIAH results are shown in Figure 1. Continuous finetuning approaches such as NTK, PI, and YaRN have successfully retrieved the "needle" within the pretraining length. Yet, only the NTK and CLEX method can retrieve the needle beyond the pretraining length, aligning with the perplexity results. The performance of the Exact

Table 2: Perplexity of different methods on PG 19 and Proof-file. NTK-32K and NTK-64K refer to NTK-Dynamic, which requires finetuning. Len refers to the longest-length examples seen at training or fine-tuning. Ex refers to the exact attention.

| | Model Details | | | Eval Length | | | | | |
|---|---|---|---|---|---|---|---|---|---|
| | Len | Ex | Methods | 2k | 4k | 8k | 16k | 32k | 64k |
| **PG19** | | | | | | | | | |
| Frozen | 4k | ✓ | LLaMA2 | **6.61** | **6.30** | - | - | - | - |
| | 4k | | LM-Infinite | **6.61** | **6.30** | 6.25 | 6.45 | 6.71 | 8.49 |
| | 4k | ✓ | NTK-Frozen | **6.61** | **6.30** | 6.82 | 7.94 | 14.52 | - |
| | 4k | | Self-Extend | **6.61** | 6.32 | 6.15 | 6.07 | 6.11 | 7.15 |
| Finetuned | 32k | ✓ | PI | 6.88 | 6.52 | 6.27 | 6.08 | 5.95 | - |
| | 32k | ✓ | NTK-32K | 6.63 | 6.32 | **6.09** | **5.92** | **5.79** | **5.76** |
| | 32k | ✓ | YaRN | 6.70 | 6.39 | 6.16 | 6.01 | 5.93 | - |
| | 32k | ✓ | CLEX | 6.85 | 6.62 | 6.14 | 5.93 | 5.82 | 5.79 |
| | 32k | | LongLora | 12.80 | 11.52 | 10.70 | 10.18 | 9.89 | - |
| | 32k | | Landmark | 8.15 | 8.14 | 8.14 | 8.11 | 8.13 | 8.15 |
| | 64k | ✓ | NTK-64K | 6.83 | 6.49 | 6.25 | 6.07 | 5.93 | 5.85 |
| **Proof-file** | | | | | | | | | |
| Frozen | 4k | ✓ | LLaMA2 | 3.34 | 3.04 | - | - | - | - |
| | 4k | | LM-Infinite | 3.34 | 3.04 | 2.94 | 3.02 | 3.11 | 3.12 |
| | 4k | ✓ | NTK-Frozen | 3.34 | 3.04 | 2.91 | 3.09 | 4.06 | 12.65 |
| | 4k | | Self-Extend | 3.35 | 3.06 | 2.88 | 2.78 | 2.75 | 2.90 |
| Finetuned | 32k | ✓ | PI | 3.34 | 3.03 | 2.83 | 2.68 | 2.58 | - |
| | 32k | ✓ | NTK-32K | **3.27** | **2.98** | **2.78** | **2.64** | **2.54** | **2.48** |
| | 32k | ✓ | YaRN | 3.29 | 3.00 | 2.81 | 2.68 | 2.59 | 106.38 |
| | 32k | ✓ | CLEX | 3.37 | 3.10 | 2.80 | 2.65 | 2.55 | **2.48** |
| | 32k | | LongLora | 5.97 | 5.10 | 4.58 | 4.27 | 4.13 | - |
| | 32k | | Landmark | 4.51 | 4.50 | 4.48 | 4.49 | 4.49 | 4.49 |
| | 64k | ✓ | NTK-64K | 3.33 | 3.03 | 2.83 | 2.69 | 2.58 | 2.51 |

Attention Method generally surpasses that of the Approximate Attention Methods. LM-Infinite and Landmark Excel are only within the local window, and they struggle to retrieve the intermediate text accurately. Regarding the Dynamic NTK method, NTK-F exhibits weak generalization when not trained. When trained on the same amount of data(1B), NTK-32K outperforms NTK-64K. When trained on 2B tokens, NTK-64K demonstrated a significant performance improvement, details are in Appendix J.

Table 3: RULER evaluation over 13 tasks at lengths from 4k to 64k.

| | **Models** | **Train Len** | **4k** | **8k** | **16k** | **32k** | **64k** | **128k** |
|---|---|---|---|---|---|---|---|---|
| Frozen | LLaMA2 | 4k | 80.94 | - | - | - | - | - |
| | LM-Infinite | 4k | 81.05 | 30.01 | 18.02 | 12.34 | 10.56 | - |
| | NTK-Frozen | 4k | 81.14 | 44.45 | 14.79 | 0.72 | 0.91 | - |
| | Self-Extend | 4k | 65.03 | 50.73 | 44.02 | 29.50 | 9.34 | - |
| Finetuned | PI | 32k | 84.56 | 76.04 | 69.64 | 57.66 | 0.00 | - |
| | NTK-32K | 32k | 86.58 | **77.75** | **70.01** | 59.42 | 46.26 | 29.91 |
| | YaRN | 32k | 79.12 | 65.60 | 54.21 | 36.95 | 0.00 | - |
| | CLEX | 32k | 50.18 | 63.93 | 64.35 | 52.17 | 30.61 | - |
| | LongLora | 32k | 10.58 | 6.37 | 3.67 | 3.53 | 0.00 | - |
| | Landmark | 32k | 22.37 | 17.52 | 16.31 | 13.56 | 14.15 | - |
| | NTK-64K | 64k | **86.60** | 76.34 | 69.56 | **60.03** | **49.31** | **40.09** |

**RULER** We test all models on 13 tasks from RULER (Hsieh et al., 2024). Each model is evaluated with 500 examples for lengths of 4k, 8k, 16k, 32k, 64k and 128k. Results are

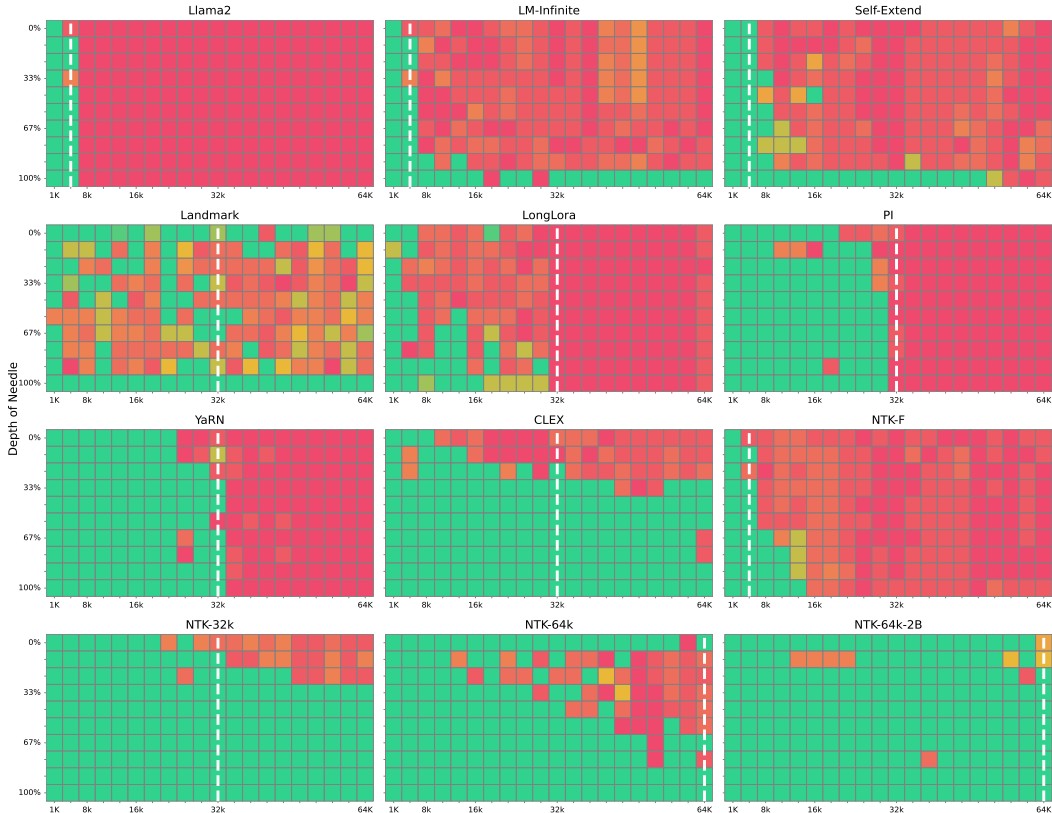

Figure 1: Needle in a Haystack evaluation. Green squares indicate a high retrieval success rate, the white dashed line denotes the longest length examples seen at training, and the Y-axis represents the distance to the retrieved target.

compared with the Llama2-7B baseline in Table 3. We observed a similar trend as in the NIHK task, NTK has the minimal performance degration w.r.t the increase of length beyond pretrained or finetuned length. NTK-32k maintained relatively good performance compared to other methods finetuned with a length cap of 32k. Performance of models on different length and breakdown by 13 subtasks can be found in Appendix M.

## 5.3 Extrinsic tasks

**Many-shot In-Context Learning with Trec News** We evaluate TREC News (Li & Roth, 2002) with 1 to 1000 in-context examples. In general, performance improves with more examples in Figure 2. Exact Attention methods show significant gains from 10 to 50 examples (+44.0%), with slower growth from 50 to 100 examples (+5.7%). Approximate Attention methods consistently underperform. Performance gains align with model perplexity; NTK-Frozen excels with fewer examples but underperforms with more.

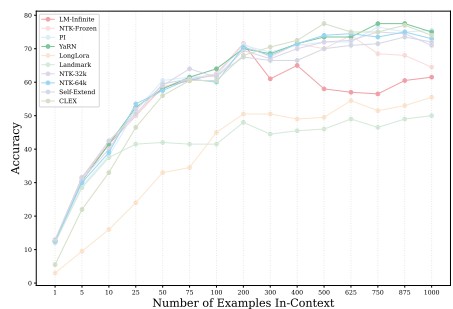

Figure 2: Many-shot ICL on TREC News.

**LongBench** Both LM-Infinite and Landmark Attention exhibit significant performance degradation compared to the base model. In contrast, the NTK, PI, and YaRN methods have successfully maintained their performance at 32k, demonstrating comparable results among these methods. PI and YaRN perform similarly in downstream tasks, while the NTK family of models remains stable.

Notably, LongLoRA also experiences a performance decline relative to the LLaMA2-base. We argue that this may be due to the sensitivity of the training procedures for LongLoRA, and we acknowledge this in our limitation discussion section. Furthermore, the overall performance on LongBench has not shown significant improvement over LLaMA2. We hypothesize that this is due to the average length of LongBench test data (7.5k) being considerably shorter than the 32k context window of the long-context methods.

Table 4: LongBench results over 16 tasks. ✓refers to exact attention.

| Exact | AvgLen | Frozen | | | | Finetuned | | | | | | |
|---|---|---|---|---|---|---|---|---|---|---|---|---|
| | | Base | Inf | N-F | SE | PI | N-32 | YaRN | CLEX | LLR | Land | N-64 |
| | | ✓ | | | ✓ | ✓ | ✓ | ✓ | | | ✓ | |
| Train Len | | 4k | 4k | 4k | 4K | 32k | 32k | 32k | 32k | 32k | 32k | 64k |
| Eval Len | | 4k | 32k | 32k | 32K | 32k | 32k | 32k | 32k | 32k | 32k | 32k |
| NQA | **18,409** | 21.09 | 10.39 | 3.88 | 23.49 | 23.02 | 23.73 | 19.82 | 24.19 | 12.07 | 12.47 | **24.31** |
| QAPR | 3,619 | 26.94 | 22.58 | 26.79 | 28.75 | 25.85 | **27.50** | 26.98 | 23.36 | 20.15 | 19.06 | 24.97 |
| MFQA | 4,559 | 32.42 | 26.19 | 29.82 | 32.66 | 35.10 | 38.22 | 37.11 | 40.83 | 24.50 | 21.86 | **40.60** |
| HPQA | 9,151 | 31.23 | 16.13 | 32.10 | 37.63 | 36.98 | **41.56** | 38.60 | 35.59 | 27.41 | 33.66 | 41.47 |
| WMQA | 4,887 | 25.75 | 20.64 | 22.34 | 30.70 | 29.38 | **31.58** | 30.63 | 28.24 | 21.46 | 24.94 | 28.62 |
| MSQ | 11,214 | 10.55 | 5.26 | 8.84 | 15.73 | 16.80 | 17.41 | **22.08** | 17.12 | 11.46 | 11.41 | 18.24 |
| GR | 8,734 | 17.32 | 13.43 | 17.87 | 13.15 | 25.61 | **28.27** | 20.98 | 24.68 | 24.05 | 17.20 | 24.37 |
| QMSM | 10,614 | 21.28 | 6.10 | 15.35 | 20.20 | 21.19 | 21.52 | 20.66 | 21.55 | 17.66 | 18.83 | **21.65** |
| MNWS | 2,113 | 3.44 | 3.63 | 9.30 | 1.50 | 10.55 | 22.13 | 8.91 | 16.96 | 21.19 | 19.43 | **25.02** |
| TREC | 5,177 | 66.00 | 61.00 | 67.50 | 69.00 | **71.00** | 69.00 | 69.00 | 67.50 | 50.00 | 49.00 | 69.00 |
| TRVQA | 8,209 | 87.89 | 81.40 | 18.69 | 88.44 | 88.55 | 88.86 | **89.63** | 89.36 | 12.28 | 74.75 | 88.65 |
| SMSM | 6,258 | 41.70 | 15.07 | 32.46 | 43.76 | **43.35** | 42.21 | 44.25 | 43.02 | 13.45 | 40.38 | 41.59 |
| PSC | 11,141 | 2.10 | 1.62 | 2.67 | 0.00 | 1.50 | 2.68 | 1.05 | 2.50 | **4.57** | 0.64 | 2.09 |
| PSR | 9,289 | **9.00** | 4.00 | 3.77 | 4.50 | 4.50 | 4.62 | 3.79 | 8.50 | 3.50 | 2.50 | 6.50 |
| LCC | 1,235 | **68.22** | 67.68 | 63.64 | 68.47 | 55.05 | 56.78 | 54.06 | 49.45 | 57.12 | 56.70 | 52.04 |
| REPO | 4,206 | **61.73** | 58.27 | 53.69 | 59.99 | 47.26 | 49.09 | 47.60 | 42.84 | 51.92 | 48.23 | 39.68 |
| Average | 7,425 | 32.92 | 25.84 | 25.54 | 33.62 | 33.48 | **35.32** | 33.45 | 33.48 | 23.30 | 28.19 | 34.30 |

**HELMET** We evaluate HELMET across lengths up to 64k tokens. The results in Table 5 shows the NTK series exhibits the least performance degradation as length increases. Detailed results breakdown by 7 task categories is provided in the Appendix A.

Table 5: HELMET evaluation over 7 categories tasks at lengths from 8k to 64k.

| | Models | Train Len | 8k | 16k | 32k | 64k |
|---|---|---|---|---|---|---|
| Frozen | NTK-Frozen | 4k | 25.81 | 16.02 | 3.46 | 1.86 |
| | Self-Extend | 4k | 27.01 | 24.40 | 19.65 | 6.69 |
| Finetuned | NTK-32k | 32k | 42.09 | 37.31 | 28.29 | 24.95 |
| | CLEX | 32k | 32.65 | 30.87 | 26.43 | 22.80 |
| | PI | 32k | 41.48 | 37.56 | 25.74 | 0.98 |
| | YaRN | 32k | 36.83 | 30.67 | 21.28 | 0.98 |
| | NTK-64k | 64k | 39.91 | 35.47 | 29.29 | 26.49 |

## 6 Additional Analysis

**Perplexity and Downstream Tasks** While prior work (Sun et al., 2021; An et al., 2023) suggests that perplexity may not reliably predict long-range task performance, our analysis in Figure 3 reveals to some extent perplexity might be reliable. We observe a general correlation between perplexity and model performance across tasks. However, we also observed that approximate attention methods, including LongLora and Landmark on RULER, show minor deviations but maintain a roughly linear relationship. We hypothesize that this apparent discrepancy with previous findings may stem from their less controlled experimental conditions and noisier datasets.

**Context extension hurts in the short term and gains in the long term** While context extension seems to improve perplexity, in Table **??**, we do not notice a significant gain in performance. We hypothesize that while this dataset contains long tasks, the average length is much shorter than 32K. These methods seem to improve the ability to model language over the long term but hurt in the short term. To understand this better we compute

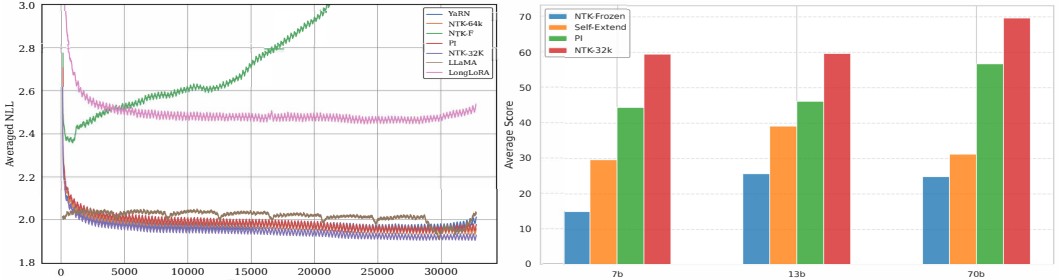

Figure 3: Perplexity and downstream task accuracy for NIAH, LongBench and RULER.

Figure 4: (Left) Average negative log-likelihood of different models by context position. (Right) Performance of different context extension methods across model sizes.

the averaged negative likelihood of each position of YaRN, LLaMa2, and NTK-32K per position (with LLaMa2 seeing just tokens every 4k chunks) in Figure 4(Left). Additionally, we evaluated these methods on short tasks from standard benchmarks and found that extension methods exhibited a slight decrease in performance on short-text tasks compared to the base model, in Table 9. This aligns with our observations in Figure 4(Left), which analyzes the average negative log-likelihood across different context positions.

**General Discoveries across Model Sizes** Our analysis across LLaMA2-7b, 13b, and 70b base models reveals several key patterns. Non-extension methods like NTK-Frozen and Self-Extend demonstrate improved performance on intrinsic tasks such as Needle-in-a-Haystack at larger scales, while maintaining consistent performance rankings across model sizes. Although continual fine-tuning methods still outperform non-extension approaches within their extension range, the correlation between perplexity and downstream task performance remains robust. These findings, shown in Figure 4(Right), provide insights into the relationship between model scaling and context extension.

**NTK Generalizes Beyond 32k** In Figure 1, we observe that NTK-32K successfully generalizes to unseen sequence lengths beyond 32k in both NIAH and RULER tasks, performing on par with NTK-64K. In contrast, NTK-F demonstrates generalization up to 8k but fails to extend further. While NTK methods may possess the capability to generalize to longer unseen sequences, their effectiveness is contingent upon conditions such as continual fine-tuning. We find that up until 4k they all improve as expected with LLaMa2 having the best NLL. After 4k they all fluctuate in average, but we see a clear separation with Yarn and NTK taking into account the long context. At extremely long context NTK remains a strong model whereas Yarn becomes reverts to a similar performance as LLaMa2.

## 7 Limitations and Conclusion

Our study has several limitations. The experiments are confined to context extensions up to 32k tokens, and behavior patterns may vary at longer extensions. Additionally, our standardized training protocol with fixed hyperparameters might disproportionately affect certain models' performance. Furthermore, our perplexity findings may be specific to our experimental settings and may not generalize to models beyond our test scope.

In this paper, we use a standardized approach to assess the performance of various long-context methods in LLMs. Our study underscores the role of perplexity as a crucial, performance indicator at length and highlights the trade-offs inherent in different attention mechanisms. We analyze the strengths and weaknesses of various approaches, providing valuable insights for future research. All our resources are open-sourced, fostering future advancements in this pivotal area of AI research.

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

# A  HELMET of LLaMA-2-7B

We present the detailed results of HELMET with LLaMA2-7B across seven subtask categories in Table X: Retrieval-Augmented Generation (RAG), Generation with Citations (Cite), Passage Re-ranking (Rerank), Many-shot In-Context Learning (ICL), Long-document Question Answering (QA), Summarization (Summ), and Synthetic Recall (Recall).

Table 6: Detailed results of HELMET with LLaMA2 variants across different evaluation lengths and task categories.

|  | Model | Recall | RAG | ICL | Cite | Rerank | QA | Summ | Avg. |
|---|---|---|---|---|---|---|---|---|---|
| NTK-Frozen | 8k | 21.31 | 50.29 | 40.60 | 4.62 | 23.25 | 25.14 | 15.47 | 25.81 |
|  | 16k | 2.38 | 39.96 | 34.36 | 4.32 | 0.87 | 16.87 | 13.41 | 16.02 |
|  | 32k | 0.00 | 2.96 | 11.56 | 0.34 | 0.00 | 3.10 | 6.23 | 3.46 |
|  | 64k | 0.00 | 0.42 | 4.76 | 0.06 | 0.00 | 3.23 | 4.59 | 1.86 |
| SelfExtend | 8k | 18.44 | 49.50 | 50.12 | 3.87 | 32.33 | 23.64 | 11.15 | 27.01 |
|  | 16k | 15.69 | 47.42 | 53.00 | 2.80 | 17.13 | 25.51 | 9.26 | 24.40 |
|  | 32k | 8.63 | 40.13 | 44.88 | 2.37 | 5.98 | 27.31 | 8.24 | 19.65 |
|  | 64k | 0.94 | 8.08 | 21.84 | 0.95 | 0.18 | 9.26 | 5.58 | 6.69 |
| NTK-32k | 8k | 72.81 | 55.46 | 61.28 | 11.32 | 45.63 | 28.47 | 19.69 | 42.09 |
|  | 16k | 57.31 | 51.08 | 66.80 | 8.35 | 27.04 | 31.38 | 19.19 | 37.31 |
|  | 32k | 29.44 | 48.42 | 69.52 | 2.75 | 6.85 | 30.36 | 10.71 | 28.29 |
|  | 64k | 17.06 | 42.79 | 72.24 | 1.95 | 0.55 | 28.87 | 11.16 | 24.95 |
| NTK-64k | 8k | 69.50 | 53.63 | 60.76 | 10.47 | 38.31 | 27.62 | 19.07 | 39.91 |
|  | 16k | 52.75 | 50.83 | 66.96 | 5.84 | 21.06 | 29.86 | 20.99 | 35.47 |
|  | 32k | 35.31 | 47.75 | 71.08 | 3.44 | 8.61 | 26.91 | 11.95 | 29.29 |
|  | 64k | 20.00 | 44.08 | 72.48 | 2.18 | 3.12 | 28.14 | 15.42 | 26.49 |
| CLEX | 8k | 44.44 | 51.00 | 50.52 | 4.29 | 37.46 | 24.40 | 16.45 | 32.65 |
|  | 16k | 45.19 | 49.21 | 58.84 | 4.35 | 18.11 | 24.24 | 16.16 | 30.87 |
|  | 32k | 28.63 | 45.25 | 68.20 | 2.06 | 0.19 | 26.52 | 14.19 | 26.43 |
|  | 64k | 15.06 | 40.04 | 72.36 | 1.45 | 0.00 | 20.76 | 9.94 | 22.80 |
| PI | 8k | 77.31 | 54.08 | 59.10 | 13.29 | 44.32 | 24.27 | 17.97 | 41.48 |
|  | 16k | 67.44 | 50.92 | 62.36 | 7.89 | 27.22 | 27.69 | 19.39 | 37.56 |
|  | 32k | 31.94 | 43.71 | 62.68 | 2.25 | 3.91 | 27.42 | 8.24 | 25.74 |
|  | 64k | 0.00 | 0.00 | 0.00 | 0.00 | 0.00 | 6.73 | 0.12 | 0.98 |
| YaRN | 8k | 61.81 | 53.21 | 58.08 | 8.83 | 29.77 | 28.67 | 17.43 | 36.83 |
|  | 16k | 35.19 | 51.08 | 63.36 | 2.69 | 21.27 | 29.13 | 12.00 | 30.67 |
|  | 32k | 14.13 | 40.88 | 61.48 | 1.28 | 1.23 | 23.53 | 6.44 | 21.28 |
|  | 64k | 0.00 | 0.00 | 0.00 | 0.00 | 0.00 | 6.67 | 0.18 | 0.98 |

# B   Results on Larger Model Sizes of LLaMA-2

## B.1   Result Overview

Result overview of 7b, 13b, and 70b models results across different extension types are shown in Table 7. Note that the perplexity is evaluated on Proof-file. Llama2-13b and Llama2-70b is evaluated on 4k context length for perplexity, Longbench and RULER.

Table 7: Overview of 7b, 13b, and 70b models results across different extension types.

| Model | Method | PPL | Needle | LongB | RULER |
|-------|--------|-----|--------|-------|-------|
| **Llama2-7b** | Base (4k) | 3.04 | 8.40 | 32.92 | 80.94 |
| | NTK-Frozen | 4.06 | 18.80 | 25.54 | 0.72 |
| | Self-Extend | 2.75 | 25.80 | 33.62 | 29.50 |
| | PI | 2.58 | 42.10 | 33.48 | 57.66 |
| | NTK-32k | 2.54 | 83.70 | 35.32 | 59.42 |
| | YaRN | 2.59 | 46.70 | 33.45 | 36.95 |
| | CLEX | 2.55 | 71.10 | 33.48 | 52.17 |
| **Llama2-13b** | Base (4k) | 2.90 | 17.00 | 33.84 | 86.35 |
| | NTK-Frozen | 3.31 | 43.00 | 31.87 | 2.30 |
| | Self-Extend | 2.65 | 53.50 | 33.69 | 30.23 |
| | PI | 2.46 | 45.00 | 37.45 | 55.95 |
| | NTK-32k | 2.44 | 82.20 | 38.41 | 58.38 |
| | YaRN | 2.46 | 44.20 | 34.03 | 44.79 |
| | CLEX | 2.43 | 78.90 | 35.89 | 52.76 |
| **Llama2-70b** | Base (4k) | 2.66 | 14.70 | 34.00 | 93.67 |
| | NTK-Frozen | 3.25 | 30.90 | 32.40 | 11.39 |
| | Self-Extend | 2.43 | 32.60 | 29.10 | 31.94 |
| | PI | 2.26 | 49.80 | 42.44 | 77.98 |
| | NTK-32k | 2.25 | 90.50 | 41.51 | 76.97 |

## B.2   Perplexity and Downstream Tasks

As shown in Figure 5, we observe a general correlation between perplexity and model performance across different model sizes. Most models exhibit a negative correlation between perplexity and performance on LongBench and RULER. However, the correlation is weaker on Needle-in-the-haystack.

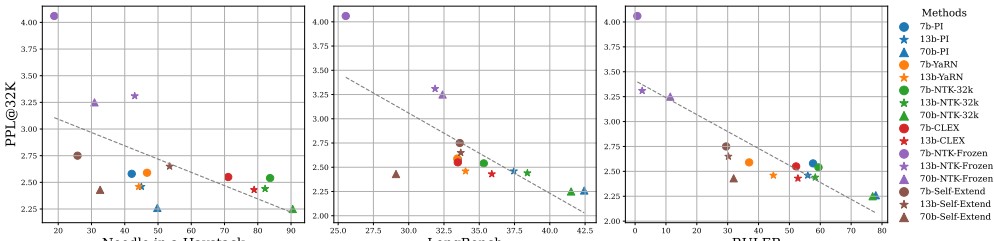

Figure 5: Perplexity and averaged downstream task accuracy for NIAH, LongBench and RULER.

## B.3   Needle-in-the-haystack

The result of the Needle-in-the-haystack task across different model sizes and extension types are shown in Figure 6. Non-extension methods like NTK-Frozen and Self-Extend demonstrate improved performance at larger scales, while maintaining consistent performance rankings across model sizes.

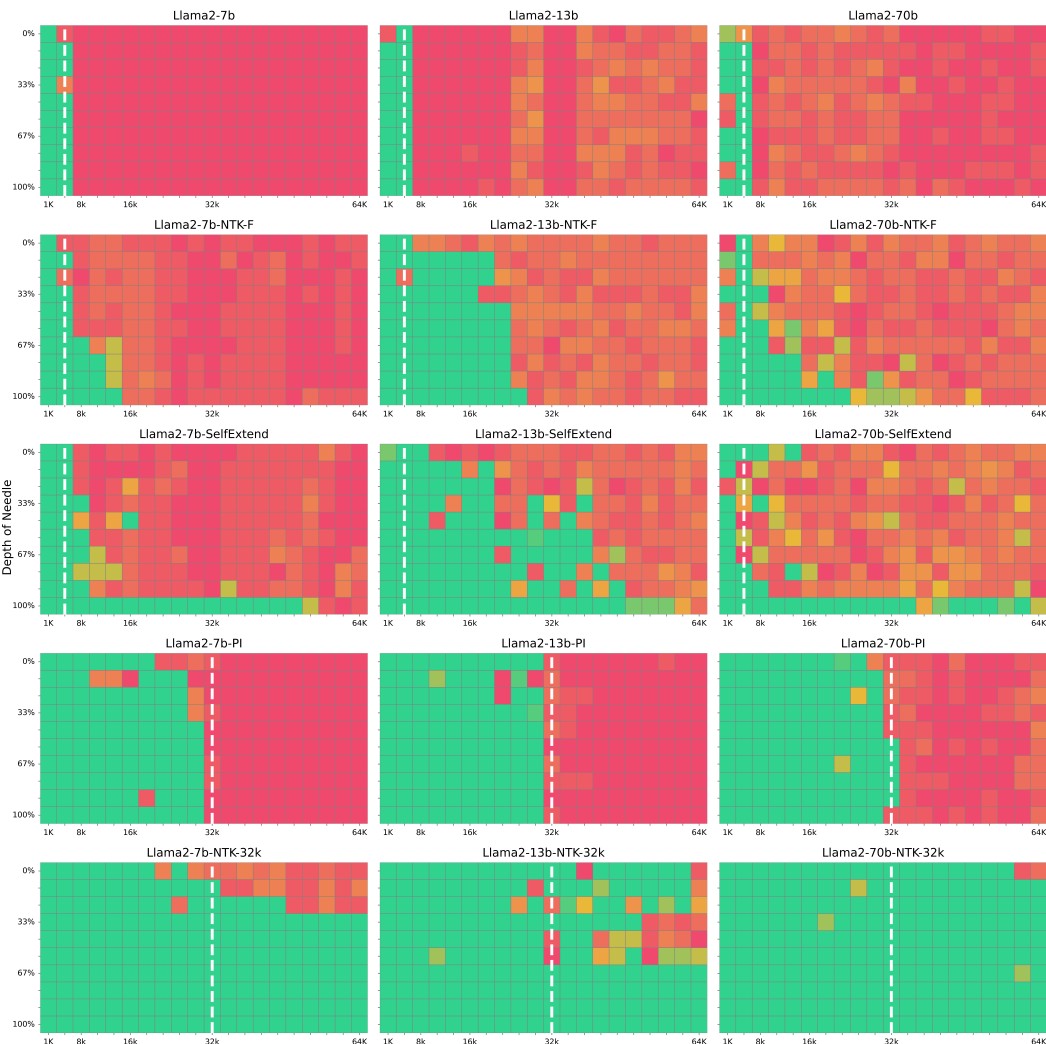

Figure 6: Needle in a Haystack evaluation with different sizes of models. Green squares indicates a high retrieval success rate, the white dashed line denotes the longest length examples seen at training or finetuning, and the Y-axis represents the distance to the retrieved target.

## C Kendall correlation of downstream task performance and perplexity

We use a non-parametric method, the ken-tau correlation to evaluate the correlation between downstream task performance and perplexity.

**Consistency Across Tasks**   The results show a strong and statistically significant negative correlation between perplexity and downstream performance for most tasks. This supports the claim that lower perplexity values are generally associated with better downstream task performance.

**Task-Specific Observations**   The strongest correlations are observed for Needle and RULER, where Kendall's tau indicates a robust alignment between perplexity and task performance rankings. For Mshots, the correlation is moderate and statistically weaker, suggesting that perplexity's predictive ability may vary slightly depending on the task.

**Impact of Perplexity Range**   Even when perplexity values are close (e.g., below 6), perplexity rankings remain a reliable indicator of downstream performance. However, the narrower range may amplify the observed performance differences, highlighting the need for nuanced interpretation.

Table 8: Perlexity and Downstream Tasks Correlations and Interpretations.

| Task | Kendall's Tau | p-value | Interpretation |
|---|---|---|---|
| Needle | -0.7191 | 0.0041 | Statistically significant ($p < 0.01$). |
| Mshots | -0.4944 | 0.0482 | Borderline significant ($p \approx 0.05$). |
| LongB | -0.6136 | 0.0149 | Statistically significant ($p < 0.05$). |
| RULER | -0.7191 | 0.0041 | Statistically significant ($p < 0.01$). |

## D Performance on short general tasks

**Short-context Tasks**   We analyzed performance on short tasks from the Open LLM Leaderboard[1] to validate our hypothesis regarding context length impact. Results are shown in Table 9. Our analysis revealed three key findings: (1) long-context extension methods generally show minor performance degradation on short-text tasks compared to the base model, with NTK-Frozen outperforming NTK-RoPE, (2) continuous fine-tuning methods demonstrate more significant short-text performance reduction, suggesting a trade-off between long and short context capabilities, and (3) these results corroborate the negative log-likelihood patterns observed in Figure 4 (Right).

Table 9: Model Performance on short Tasks. HS refers to Hellaswag, TQA refers to TruthfulQA and WG refers to WinoGrande.

| Methods | ARC-c | ARC-e | HS | MMLU | TQA | WG | Avg. |
|---|---|---|---|---|---|---|---|
| Llama2-7b | 52.73 | 81.31 | 78.96 | 42.09 | 38.97 | 74.43 | 61.42 |
| LM-Infinite | 52.56 | 81.36 | 78.95 | 42.09 | 38.96 | 74.11 | 61.34 |
| Self-Extend | 52.56 | 81.31 | 78.94 | 42.07 | 38.97 | 74.43 | 61.38 |
| NTK-Frozen | 52.73 | 81.31 | 78.96 | 42.09 | 38.97 | 74.43 | 61.42 |
| PI | 51.11 | 81.14 | 77.44 | 37.19 | 38.03 | 71.74 | 59.44 |
| NTK-32k | 49.15 | 80.22 | 74.48 | 35.25 | 38.13 | 72.61 | 58.31 |
| NTK-64k | 46.08 | 78.32 | 70.68 | 34.27 | 39.08 | 70.24 | 56.45 |
| YaRN | 53.41 | 81.82 | 78.47 | 41.06 | 38.63 | 74.43 | 61.30 |
| CLEX | 50.60 | 81.27 | 76.06 | 37.54 | 36.10 | 64.72 | 57.72 |
| LongLora | 46.67 | 78.58 | 67.08 | 26.29 | 37.61 | 55.25 | 51.91 |

---

[1]https://huggingface.co/spaces/open-llm-leaderboard-old/open_llm_leaderboard

# E   Efficiency Analysis

We conduct inference speed comparisons under controlled conditions using the same hardware setup. As shown in Table 10, we observed that approximate attention methods are indeed faster, achieving a speedup of approximately 1.5x to 2x compared to LLaMA when the context length is short; however, when the context length gets longer, we didn't see a significant margin. We hypothesize that the discrepancy between the theoretical FLOPs-based comparisons and the observed speedup arises due to differences in hardware characteristics and CUDA implementations of the respective methods.

Table 10: Efficiency analysis for different sequence lengths (4k, 8k, 16k, 32k). The prefill time(Pre) cost represents the time required to generate the first token. The decoding speed(Dec) (seconds / per token) is averaged over 100 token inferences at each sequence length. Memory consumption corresponds to the peak GPU memory usage during inference. All methods, except for LM-Infinite and Landmark, utilize Flash-Attention 2 for enhanced computational efficiency.

| Method | 4k | | | 8k | | | 16k | | | 32k | | |
|---|---|---|---|---|---|---|---|---|---|---|---|---|
| | Pre (s) | Dec (s) | Mem (GB) | Pre (s) | Dec (s) | Mem (GB) | Pre (s) | Dec (s) | Mem (GB) | Pre (s) | Dec (s) | Mem (GB) |
| Llama2 | 1.15 | 0.03 | 17.13 | 1.51 | 0.06 | 21.61 | 2.41 | 0.11 | 30.59 | 4.63 | 0.21 | 48.55 |
| NTK-F | 1.16 | 0.04 | 17.13 | 1.56 | 0.05 | 21.61 | 2.39 | 0.06 | 30.59 | 4.69 | 0.09 | 48.55 |
| PI | 1.15 | 0.03 | 22.05 | 1.54 | 0.03 | 26.54 | 2.43 | 0.05 | 35.51 | 4.74 | 0.08 | 53.47 |
| NTK-32k | 1.17 | 0.04 | 17.11 | 1.56 | 0.04 | 21.60 | 2.42 | 0.06 | 30.58 | 4.75 | 0.09 | 48.53 |
| YaRN | 1.23 | 0.03 | 18.05 | 1.53 | 0.03 | 22.54 | 2.43 | 0.05 | 31.51 | 4.80 | 0.08 | 49.47 |
| CLEX | 1.16 | 0.05 | 17.16 | 6.99 | 0.07 | 21.74 | 7.68 | 0.11 | 30.92 | 10.06 | 0.18 | 49.28 |
| LM-Infinite | 1.56 | 0.05 | 17.23 | 3.34 | 0.07 | 25.47 | 5.82 | 0.11 | 38.60 | 11.58 | 0.18 | 65.61 |
| Self-Extend | 1.24 | 0.05 | 17.23 | 1.63 | 0.07 | 21.81 | 2.63 | 0.13 | 30.98 | 4.97 | 0.22 | 49.32 |
| LongLora | 1.16 | 0.05 | 17.16 | 1.65 | 0.05 | 21.65 | 2.60 | 0.05 | 30.62 | 5.07 | 0.08 | 48.58 |
| Landmark | 8.62 | 0.08 | 18.77 | 17.65 | 0.08 | 22.97 | 36.47 | 0.09 | 31.22 | 77.77 | 0.09 | 47.74 |

# F   Details of Yarn Extension Method

**YaRN**, another RoPE extension method, uses "NTK-by-parts" interpolation strategies across different dimensions of the embedding space and introduces a temperature factor to adjust the attention distribution for long inputs.

$$\alpha_j^{\text{YaRN}} = ((1 - \gamma_j) \frac{1}{t} + \gamma_j) / \sqrt{T} \tag{14}$$

We use a ramp vector $\gamma$ to determine the interpolation between the $\frac{1}{t}$ and the original frequency base. The interpolation gating is set based on the frequency for the dimension $j$.

$$\gamma_j = \begin{cases} 0, & \text{if } \theta_j < p, \\ 1, & \text{if } \theta_j > q, \\ \frac{\theta_j - p}{q - p}, & \text{otherwise.} \end{cases} \tag{15}$$

The values of $p, q, T$ can be tuned as needed.

## G    LLaMA-3 and Phi-2 for Context Extension

We use other open-weight models, LLaMA-3-8B (AI@Meta, 2024) base and Phi-2-base (Java-heripi et al., 2023) as the base point for context extension, to verify whether the trends and analyses we observed are consistent across different base models. Using an identical training recipe, we re-train and re-evaluate seven models with Llama-3-8B base and Phi-2-base.

### G.1    Perplexity on Proof-file of Llama-3 and Phi-2

We evaluate the perplexity of LLaMA-3-8B base in Table 11 and Phi-2-base in Table 12. Consistent with our observations on LLaMA-2-7B, continuous fine-tuning methods like PI and YaRN effectively maintain low perplexity scores within the pre-training context length. However, perplexity scores escalate once the context length exceeds the pre-trained window. Notably, only NTK and CLEX could generalize to unseen sequence lengths during both pre-training and continual fine-tuning.

Table 11: Perplexity results of different methods on Proof-file with LLaMA-3-8B base. Len refers to the longest-length examples seen at training or fine-tuning. Ex refers to the exact attention. All results are produced by our experiments.

|  | \multicolumn{3}{c}{Model Details} | \multicolumn{6}{c}{Eval Length} |
|  | Len | Ex | Methods | 2k | 4k | 8k | 16k | 32k | 64k |
|---|---|---|---|---|---|---|---|---|---|
| Frozen | 8k | ✓ | LLaMA-3 | **2.98** | **2.72** | **2.54** | 31.11 | 318 | - |
|  | 8k | ✓ | NTK-Frozen | **2.98** | **2.72** | **2.54** | 2.48 | 3.80 | 8.69 |
|  | 8k |  | Self-Extend | **2.98** | **2.72** | **2.54** | **2.43** | 2.36 | 2.62 |
| Finetuned | 32k | ✓ | PI | 3.13 | 2.83 | 2.63 | 2.49 | 2.39 | 38.77 |
|  | 32k | ✓ | NTK-32K | 3.05 | 2.76 | 2.57 | **2.43** | **2.34** | **2.28** |
|  | 32k | ✓ | YaRN | 3.16 | 2.86 | 2.66 | 2.52 | 2.43 | 4989 |
|  | 32k | ✓ | CLEX | 3.30 | 2.89 | 2.64 | 2.47 | 2.38 | 2.39 |

Table 12: Perplexity results of different methods on Proof-file with Phi-2-base. Len refers to the longest-length examples seen at training or fine-tuning. Ex refers to the exact attention. All results are produced by our experiments.

|  | \multicolumn{3}{c}{Model Details} | \multicolumn{6}{c}{Eval Length} |
|  | Len | Ex | Methods | 2k | 4k | 8k | 16k | 32k | 64k |
|---|---|---|---|---|---|---|---|---|---|
| Frozen | 2k | ✓ | Phi-2-base | **4.02** | 25.72 | 175.05 | - | - | - |
|  | 2k | ✓ | NTK-Frozen | **4.02** | 3.73 | 4.07 | 5.49 | 12.58 | 36.68 |
|  | 2k |  | Self-Extend | 4.08 | **3.70** | **3.48** | 3.42 | 3.48 | 3.73 |
| Finetuned | 32k | ✓ | PI | 7.53 | 6.75 | 6.25 | 5.97 | 5.83 | 45.00 |
|  | 32k | ✓ | NTK-32K | 4.24 | 3.81 | 3.51 | **3.32** | **3.18** | **3.20** |
|  | 32k | ✓ | CLEX | 5.53 | 4.32 | 3.78 | 3.51 | 3.42 | 3.60 |
|  | 64k | ✓ | NTK-64K | 4.63 | 4.14 | 3.82 | 3.61 | 3.47 | 3.38 |

### G.2    RULER of LLaMA-3 and Phi-2

We test all models on all 13 diverse tasks for LLaMA-3-8B and 12 tasks (except QA-2) for Phi-2 from the four Ruler Hsieh et al. (2024) categories in Table 13 and Table 14. Consistently, NTK-32k maintains relatively strong performance compared to other methods fine-tuned with a length cap of 32k and showing a slight drop in performance at 64k. The only exception is Self-Extend on LLaMA-3-8B, which benefits from a larger pretraining length of 8192. Self-Extend demonstrates superior performance on LLaMA-3 compared to LLaMA-2 and Phi-2, with performance approaching that of CLEX and PI.

Table 13: RULER evaluation on seven methods with LLaMA-3-8B. Performance of models evaluated at length from 8k to 64k. Each score is computed by averaging the accuracy of 13 tasks. Train Len refers to the longest-length examples seen at continuous finetuning.

| | Models | Train Len | 4k | 8k | 16k | 32k | 64k |
|---|---|---|---|---|---|---|---|
| Frozen | LLaMA-3 | 8k | 93.63 | 91.16 | 0.06 | 0.01 | 0.05 |
| | NTK-Frozen | 8k | 93.63 | 91.15 | 6.86 | 1.98 | 0.02 |
| | Self-Extend | 8k | 92.73 | 84.11 | 78.78 | 71.40 | 38.79 |
| Finetuned | PI | 32k | 91.60 | 88.56 | 86.99 | 73.14 | 0.02 |
| | NTK-32K | 32k | **93.68** | **91.67** | **91.12** | **86.04** | **65.42** |
| | YaRN | 32k | 92.51 | 90.59 | 88.07 | 68.69 | 0.06 |
| | CLEX | 32k | 89.65 | 87.35 | 87.89 | 69.27 | 39.81 |

Table 14: RULER evaluation on seven methods with Phi-2-base. Performance of models evaluated at length from 2k to 64k. Each score is computed by averaging the accuracy of 12 tasks. Train Len refers to the longest-length examples seen at continuous finetuning.

| | Models | Train Len | 2k | 4k | 8k | 16k | 32k | 64k |
|---|---|---|---|---|---|---|---|---|
| Frozen | Phi-2-base | 2k | 83.73 | - | - | - | - | - |
| | NTK-Frozen | 2k | **83.98** | 52.95 | 18.09 | 4.07 | 0.06 | 0.00 |
| | Self-Extend | 2k | 68.55 | 50.82 | 36.65 | 22.00 | 7.83 | 2.32 |
| Finetuned | PI | 32k | 25.51 | 23.19 | 16.88 | 14.99 | 4.78 | 0.00 |
| | NTK-32K | 32k | 81.18 | 66.90 | 52.57 | **46.53** | **32.06** | 12.84 |
| | CLEX | 32k | 75.33 | **72.66** | **53.56** | 46.23 | 25.46 | 13.03 |
| | NTK-64K | 64k | 78.73 | 59.87 | 47.56 | 41.87 | 25.66 | **17.69** |

## G.3 HELMET of LLaMA-3

We evaluate HELMET on all 7 categories of tasks across lengths up to 128k tokens on LLaMA-3-8B in Table 15.

Table 15: HELMET evaluation on LLaMA-3-8B at lengths from 8k to 128k.

| | Models | Train Len | 8k | 16k | 32k | 64k | 128k |
|---|---|---|---|---|---|---|---|
| Frozen | NTK-Frozen | 4k | 47.48 | 38.98 | 3.16 | 2.98 | 2.11 |
| | Self-Extend | 4k | 44.28 | 41.85 | 38.59 | 27.90 | 10.41 |
| Finetuned | NTK-32k | 32k | 50.51 | 48.96 | 47.14 | 37.08 | 19.95 |
| | CLEX | 32k | 46.68 | 47.14 | 42.57 | 29.43 | 17.41 |
| | PI | 32k | 49.22 | 47.66 | 45.78 | 2.43 | 1.56 |
| | YaRN | 32k | 48.47 | 48.65 | 45.31 | 2.95 | 1.77 |
| | NTK-64k | 64k | 49.86 | 49.15 | 47.68 | 42.91 | 37.11 |

## H  Implementation Details

### H.1  Training

To maintain consistency across all models, we use a fixed training protocol (Chen et al., 2023b). We adopt standard practices by applying an exponential moving average (EMA) to the model weights with a constant decay rate. Most training hyperparameters are based on (Fu et al., 2024), including a learning rate of $2 \times 10^{-5}$. We implement a linear warm-up for the learning rate and set the weight decay to zero, utilizing 8 NVIDIA A100 GPUs. We

present the hyperparameter settings for different methods on LLaMA-2-7B, LLaMA-3-8B, and Phi-2 during the training stage in Table 16.

**LLaMA-2**    For LongLora, we fine-tune the LoRA adapter weights along with trainable embeddings and normalization, subsequently integrating these trained weights into the LLaMA2 base model for evaluation. For Landmark Attention, the training context length is 512, with a block size of 64. For YaRN, we set beta fast to 32, beta slow to 1, and $\alpha$ to 8.0. For CLEX, we set the max scale factor to 32 and use the SiLU activation function. For NTK-RoPE, given the maximum observed length during training or inference, $C_{\text{test}}$, and the scaling hyperparameter $s$, we follow Fu et al. (2024) in replacing $t$ with $s \cdot \frac{\max(C', C_{\text{test}})}{C} - (s - 1)$, and set the hyperparameter $s$ to $\frac{C'}{2C}$ during both training and inference. We set $s$ to 4.0 for NTK-32k and $s$ to 8.0 for NTK-64k. For LM-infinite, we set the global memory $G = 10$ and the local window $M = 4096$.

We reuse the original scale factor to maintain consistency for NTK, YaRN, and Position Interpolation methods. However, this base factor significantly degrades continual fine-tuned models, particularly causing performance deterioration in shorter sequences. Therefore, we conduct a grid search to determine a better scale factor for different input lengths for NTK-RoPE method. Based on our findings, we follow and improve upon Fu et al. (2024) to set the scale factor for NTK-RoPE method. The scale factor and its relationship with perplexity are reported in the Appendix K.

**LLaMA-3**    For YaRN, we set beta fast to 32, beta slow to 1, and $\alpha$ to 4.0. For CLEX, we set the max scale factor to 16 and use the SiLU activation function. For NTK-RoPE, we set $s$ to 2.0.

**Phi-2**    For CLEX, we set the max scale factor to 64 and use the tanh activation function. For NTK-RoPE, we set $s$ to 8.0 for NTK-32k and 16.0 for NTK-64k.

Table 16: Hyperparameters for Different Long Sequence Methods in Training.

| Models | Methods | Train Len | Train Tokens | $\alpha$ | bsz | lr |
|---|---|---|---|---|---|---|
| | PI | 32k | 1B | 8.0 | 32 | 2e-5 |
| | NTK-32K | 32k | 1B | 29.0 | 32 | 2e-5 |
| | YaRN | 32k | 1B | 8.0 | 32 | 2e-5 |
| LLaMA-2 | LongLora | 32k | 1B | 8.0 | 32 | 2e-5 |
| | Landmark | 32k | 1B | - | 32 | 2e-5 |
| | NTK-64K | 64k | 1B | 57.0 | 32 | 2e-5 |
| | NTK-64K-2B | 64k | 2B | 57.0 | 32 | 2e-5 |
| | PI | 32k | 1B | 4.0 | 32 | 2e-5 |
| LLaMA-3 | NTK-32K | 32k | 1B | 7.0 | 32 | 2e-5 |
| | YaRN | 32k | 1B | 4.0 | 32 | 2e-5 |
| | PI | 32k | 1B | 16.0 | 32 | 2e-5 |
| Phi-2 | NTK-32K | 32k | 1B | 121.0 | 32 | 2e-5 |
| | NTK-64K | 64k | 1B | 497.0 | 32 | 2e-5 |

## H.2    Inference

For all methods on all base models, we show the hyperparameter settings and present the $\alpha$ used for different length ranges during inference in Table 17.

**LLaMA-2**    For Landmark Attention, the training context length is set to 512, with a block size of 64. For Self-Extend, we set the local window size $M$ for neighbor tokens to 1024 and the group size $N$ to 64. For NTK-RoPE, we replace $t$ with $s \cdot \frac{\max(C', C_{\text{test}})}{C} - (s - 1)$ and set $s$ to 4.0 for NTK-32k and 8.0 for NTK-64k.

**LLaMA-3** For Self-Extend, we set the local window size $M$ for neighbor tokens to 2048 and the group size $N$ to 32. For NTK-RoPE, we set $s$ to 2.0 for NTK-frozen and 4.0 NTK-32k.

**Phi-2** For Self-Extend, we set the local window size $M$ for neighbor tokens to 512 and the group size $N$ to 128. For NTK-RoPE, we set $s$ to 2.0 for NTK-frozen, 8.0 for NTK-32k, and 16.0 for NTK-64k.

Table 17: Hyperparameters for the Scale Factor $\alpha$ Different Long-context Methods in Inference.

| Models | Methods | 4k | 8k | 16k | 32k | 64k |
|---|---|---|---|---|---|---|
| LLaMA-2 | NTK-Frozen | 1.0 | 3.0 | 7.0 | 15.0 | 31.0 |
| | PI | 8.0 | 8.0 | 8.0 | 8.0 | 8.0 |
| | NTK-32K | 29.0 | 29.0 | 29.0 | 29.0 | 61.0 |
| | YaRN | 8.0 | 8.0 | 8.0 | 8.0 | 8.0 |
| | LongLora | 8.0 | 8.0 | 8.0 | 8.0 | 8.0 |
| | NTK-64K | 57.0 | 57.0 | 57.0 | 57.0 | 57.0 |
| LLaMA-3 | NTK-Frozen | 1.0 | 1.0 | 3.0 | 7.0 | 15.0 |
| | PI | 4.0 | 4.0 | 4.0 | 4.0 | 4.0 |
| | NTK-32K | 13.0 | 13.0 | 13.0 | 13.0 | 29.0 |
| | YaRN | 4.0 | 4.0 | 4.0 | 4.0 | 4.0 |
| Phi-2 | NTK-Frozen | 3.0 | 7.0 | 15.0 | 31.0 | 63.0 |
| | PI | 16.0 | 16.0 | 16.0 | 16.0 | 16.0 |
| | NTK-32K | 121 | 121 | 121 | 121 | 249 |
| | NTK-64K | 497 | 497 | 497 | 497 | 497 |

## I Training Data Construction

We sample 1B tokens from a long-context data mixture following Fu et al. (2024). We use the SlimPajama (Soboleva et al., 2023) dataset for continuous finetuning. This dataset serves as an open-source replication of the LLaMA (Touvron et al., 2023) pretraining data mixture. It comprises 82% web data (sourced 67% from CommonCrawl and 15% from C4), 4.5% code data (Github), 4.5% Wikipedia content, 4.5% books, 2.5% Arxiv papers, and 2.0% StackExchange content. We use per-source length-upsampling to sample 1B tokens from the datasets, which increases the portion of long sequences while keeping the domain mixture the same. We packed all sampled data into chunks of the corresponding training length, regardless of document boundaries, following common practiceTouvron et al. (2023); Fu et al. (2024).

## J Longer Model Needs more Training Tokens

We observe that the performance of NTK-64K is not as good as NTK-32K. Consequently, we further sample 2B tokens from a long-context data mixture from Fu et al. (2024) for training and evaluate the model on the "Needle in A Haystack" task, as shown in Figure 7. Our NTK-64K model demonstrates a significant performance improvement when trained with more tokens, indicating that longer models require more tokens for effective training.

## K RoPE Scale Factor for Dynamic NTK

We observe that the scale factor significantly degrades NTK-Dynamic models, particularly causing performance deterioration in shorter sequences. Therefore, we conduct a grid search to determine a better scale factor for different input lengths. The scale factor and its relationship with perplexity on PG19 are reported in Table 18.

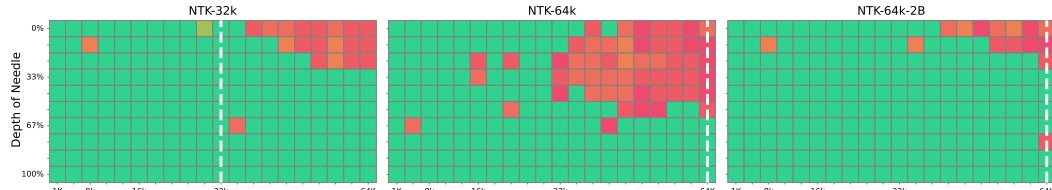

Figure 7: Needle in a Haystack evaluation. "NTK-64-2B" represents the NTK-64K model trained with 2B tokens. Green squares indicates a high retrieval success rate, the white dashed line denotes the longest length examples seen at training or finetuning, and the Y-axis represents the distance to the retrieved target.

Table 18: The scale factor and its relationship with perplexity on PG19. We only use the first 2 documents of PG19 to calculate the perplexity.

| Models | Scale Factor | 4k | 8k | 16k | 32k | 64k |
|---|---|---|---|---|---|---|
| NTK-Frozen | 1 | **7.65** | 118.82 | NaN | NaN | NaN |
| | 3 | 8.19 | **7.99** | 57.15 | 386.02 | NaN |
| | 7 | 9.39 | 9.26 | **9.61** | 72.62 | 486.13 |
| | 15 | 11.53 | 12.04 | 12.98 | **20.15** | 180.59 |
| | 31 | 16.18 | 20.66 | 26.67 | 40.06 | **69.01** |
| | 63 | 30.22 | 48.78 | 69.89 | 89.75 | 118.59 |
| NTK-32K | 1 | 12.64 | NaN | NaN | NaN | NaN |
| | 5 | 7.84 | 7.638 | 10.36 | NaN | NaN |
| | 13 | **7.686** | 7.459 | 7.25 | 8.35 | NaN |
| | 29 | 7.689 | **7.457** | **7.24** | **6.82** | 9.11 |
| | 61 | 7.8 | 7.565 | 7.34 | 6.91 | **6.63** |
| | 125 | 7.99 | 7.774 | 7.57 | 7.13 | 6.83 |
| NTK-64K | 1 | 19.16 | NaN | NaN | NaN | NaN |
| | 9 | 8.02 | 7.79 | 7.63 | 22.6 | NaN |
| | 25 | **7.89** | **7.65** | **7.443** | 7.04 | 14.02 |
| | 57 | 7.922 | 7.67 | 7.44 | **7.01** | **6.75** |
| | 121 | 8.016 | 7.75 | 7.51 | 7.06 | 6.77 |

## L  LongLora Validation

To validate our LongLora Chen et al. (2023b) implementation, we reproduce their Llama-2-7b-longlora-32k model following LongLora's training data and training recipe. We evaluate the perplexity for the corresponding length on PG19 and Proof-file in Table 19.

Table 19: Perplexity results of LongLora reported and our reproduction on PG 19 and Proof-file.

| Method | 2k | 4k | 8k | 16k | 32k |
|---|---|---|---|---|---|
| **PG19** | | | | | |
| Llama-2-7b-longlora-32k | 8.29 | 7.83 | 7.54 | 7.35 | 7.22 |
| Our Reproduction | 8.10 | 7.69 | 7.43 | 7.28 | 7.32 |
| **Proof-file** | | | | | |
| Llama-2-7b-longlora-32k | 3.35 | 3.01 | 2.78 | 2.61 | 2.50 |
| Our Reproduction | 3.33 | 3.01 | 2.80 | 2.67 | 2.61 |

## M  RULER Subtasks Result

The performance of models on different lengths and breakdowns by 13 subtasks are reported in Table 20(RULER on 4k), Table 21(RULER on 8k), Table 22(RULER on 16k), Table 23(RULER on 32k) and Table 24(RULER on 64k).

Table 20: Ruler results on 4k context length. N-32 and N-64 refer to NTK finetuned on 32K and 64K context lengths respectively. Inf refers to LM-Infinite. SE refers to Self-Extend. LLR refers to LongLora. Train Len refers to the longest length examples seen at training or finetuning. Eval Len refers to the maximum length of the input prompt. ✓refers to whether the method is exact attention.

| Exact | Frozen | | | | Finetuned | | | | | | |
|---|---|---|---|---|---|---|---|---|---|---|---|
| | Base ✓ | Inf | N-F ✓ | SE | PI ✓ | N-32 ✓ | YaRN ✓ | CLEX ✓ | LLR | Land | N-64 ✓ |
| Train Len | 4k | 4k | 4k | 4k | 32k | 32k | 32k | 32k | 32k | 32k | 64k |
| Eval Len | 4k | 4k | 4k | 4k | 4k | 4k | 4k | 4k | 4k | 4k | 4k |
| NIAH_S1 | 100.00 | 100.00 | 100.00 | 100.00 | 98.00 | 100.00 | 99.60 | 100.00 | 0.00 | 49.00 | 100.00 |
| NIAH_S2 | 100.00 | 100.00 | 100.00 | 100.00 | 99.80 | 100.00 | 88.60 | 100.00 | 0.00 | 20.60 | 100.00 |
| NIAH_S3 | 99.20 | 95.80 | 98.80 | 89.80 | 99.80 | 94.20 | 53.00 | 89.60 | 0.00 | 10.00 | 97.20 |
| NIAH_M1 | 99.20 | 98.80 | 99.20 | 79.00 | 99.20 | 99.20 | 62.60 | 95.80 | 0.00 | 10.60 | 98.00 |
| NIAH_M2 | 88.00 | 88.00 | 88.20 | 26.00 | 95.40 | 97.40 | 14.00 | 83.20 | 0.00 | 6.80 | 97.00 |
| NIAH_M3 | 61.40 | 62.00 | 61.60 | 14.40 | 78.00 | 68.20 | 8.20 | 53.80 | 0.00 | 1.20 | 84.80 |
| NIAH_MV | 83.55 | 90.45 | 86.60 | 82.10 | 95.45 | 96.40 | 50.25 | 95.10 | 0.05 | 10.80 | 96.15 |
| NIAH_MQ | 95.45 | 96.15 | 96.00 | 90.70 | 96.95 | 97.00 | 62.00 | 96.20 | 0.00 | 5.35 | 98.25 |
| VT | 57.72 | 58.56 | 56.48 | 8.92 | 96.64 | 98.16 | 25.68 | 85.72 | 0.00 | 2.92 | 97.00 |
| CWE | 78.20 | 75.90 | 78.20 | 73.56 | 81.38 | 80.86 | 58.78 | 82.60 | 64.70 | 23.16 | 74.26 |
| FWE | 84.33 | 84.20 | 84.93 | 80.07 | 58.40 | 85.53 | 26.20 | 52.60 | 18.53 | 84.93 | 81.40 |
| QA_1 | 62.20 | 60.40 | 62.40 | 60.60 | 57.80 | 62.40 | 60.20 | 55.80 | 26.20 | 37.20 | 55.80 |
| QA_2 | 43.00 | 43.40 | 42.40 | 40.20 | 42.40 | 46.20 | 43.20 | 38.20 | 28.00 | 28.20 | 46.00 |
| Avg. | 80.94 | 81.05 | 81.14 | 65.03 | 84.56 | 86.58 | 50.18 | 79.12 | 10.58 | 22.37 | 86.60 |

Table 21: Ruler results on 8k context length.

| Exact | Frozen | | | | Finetuned | | | | | | |
|---|---|---|---|---|---|---|---|---|---|---|---|
| | Base ✓ | Inf | N-F ✓ | SE | PI ✓ | N-32 ✓ | YaRN ✓ | CLEX ✓ | LLR | Land | N-64 ✓ |
| Train Len | 4k | 4k | 4k | 4k | 32k | 32k | 32k | 32k | 32k | 32k | 64k |
| Eval Len | 4k | 4k | 4k | 4k | 4k | 4k | 4k | 4k | 4k | 4k | 4k |
| NIAH_S1 | - | 46.00 | 61.60 | 100.00 | 99.00 | 99.80 | 100.00 | 100.00 | 0.00 | 46.00 | 100.00 |
| NIAH_S2 | - | 36.60 | 59.40 | 98.80 | 100.00 | 100.00 | 99.40 | 100.00 | 0.00 | 7.20 | 100.00 |
| NIAH_S3 | - | 20.80 | 51.00 | 88.60 | 99.20 | 94.20 | 96.00 | 97.80 | 0.00 | 3.80 | 99.20 |
| NIAH_M1 | - | 27.80 | 46.00 | 69.40 | 98.00 | 94.20 | 86.60 | 90.20 | 0.00 | 7.60 | 95.20 |
| NIAH_M2 | - | 4.40 | 11.00 | 8.20 | 91.60 | 86.20 | 60.60 | 66.00 | 0.00 | 1.60 | 86.60 |
| NIAH_M3 | - | 2.60 | 4.00 | 3.20 | 48.40 | 52.20 | 34.60 | 11.80 | 0.00 | 0.00 | 47.40 |
| NIAH_MV | - | 30.35 | 41.35 | 52.95 | 65.50 | 85.95 | 70.40 | 61.25 | 0.00 | 6.25 | 84.75 |
| NIAH_MQ | - | 30.15 | 50.40 | 78.70 | 93.25 | 95.20 | 92.45 | 86.95 | 0.00 | 3.35 | 94.95 |
| VT | - | 4.88 | 69.88 | 1.48 | 91.20 | 96.16 | 77.52 | 48.16 | 0.00 | 3.08 | 94.36 |
| CWE | - | 65.08 | 40.30 | 30.82 | 45.66 | 45.76 | 44.72 | 32.72 | 18.92 | 22.08 | 40.80 |
| FWE | - | 56.73 | 64.87 | 59.00 | 65.07 | 70.13 | 10.53 | 45.40 | 16.73 | 76.60 | 54.13 |
| QA_1 | - | 35.80 | 44.40 | 31.00 | 50.80 | 49.20 | 43.20 | 48.20 | 22.80 | 25.00 | 50.20 |
| QA_2 | - | 29.00 | 33.60 | 37.40 | 40.80 | 41.80 | 36.80 | 42.60 | 24.40 | 25.20 | 44.80 |
| Avg. | - | 30.01 | 44.45 | 50.73 | 76.04 | 77.75 | 65.60 | 63.93 | 6.37 | 17.52 | 76.34 |

Table 22: Ruler results on 16k context length.

| Exact | | Frozen | | | | | | Finetuned | | | |
|---|---|---|---|---|---|---|---|---|---|---|---|
| | Base ✓ | Inf | N-F ✓ | SE | PI ✓ | N-32 ✓ | YaRN ✓ | CLEX ✓ | LLR | Land | N-64 ✓ |
| Train Len | 4k | 4k | 4k | 4k | 32k | 32k | 32k | 32k | 32k | 32k | 64k |
| Eval Len | 4k | 4k | 4k | 4k | 4k | 4k | 4k | 4k | 4k | 4k | 4k |
| NIAH_S1 | - | 21.00 | 14.20 | 99.80 | 97.20 | 99.40 | 100.00 | 99.80 | 0.00 | 42.40 | 99.80 |
| NIAH_S2 | - | 17.00 | 17.40 | 93.40 | 100.00 | 100.00 | 99.20 | 100.00 | 0.20 | 6.80 | 100.00 |
| NIAH_S3 | - | 11.60 | 8.20 | 77.00 | 99.60 | 98.60 | 89.60 | 99.60 | 0.00 | 3.60 | 100.00 |
| NIAH_M1 | - | 15.80 | 9.20 | 60.00 | 97.80 | 93.20 | 83.40 | 89.40 | 0.00 | 5.60 | 90.80 |
| NIAH_M2 | - | 0.00 | 0.60 | 3.80 | 82.80 | 79.80 | 19.60 | 72.00 | 0.00 | 0.80 | 67.60 |
| NIAH_M3 | - | 1.00 | 0.00 | 1.80 | 34.20 | 18.20 | 7.40 | 15.00 | 0.00 | 0.00 | 29.60 |
| NIAH_MV | - | 8.40 | 6.90 | 38.85 | 77.55 | 81.95 | 58.75 | 62.40 | 0.00 | 4.80 | 83.50 |
| NIAH_MQ | - | 8.85 | 7.95 | 59.30 | 90.95 | 86.20 | 85.15 | 81.60 | 0.00 | 2.75 | 90.35 |
| VT | - | 6.56 | 11.28 | 1.16 | 68.84 | 83.56 | 47.12 | 48.16 | 0.00 | 2.52 | 88.68 |
| CWE | - | 19.94 | 28.36 | 17.80 | 27.26 | 26.32 | 23.72 | 28.60 | 0.62 | 11.90 | 21.20 |
| FWE | - | 77.13 | 25.80 | 59.80 | 47.93 | 61.73 | 10.13 | 57.33 | 12.93 | 81.60 | 51.73 |
| QA_1 | - | 22.80 | 36.40 | 28.00 | 46.00 | 45.20 | 43.20 | 49.20 | 13.20 | 23.00 | 45.00 |
| QA_2 | - | 24.20 | 26.00 | 31.60 | 35.20 | 36.00 | 37.40 | 33.40 | 20.80 | 26.20 | 36.00 |
| Avg. | - | 18.02 | 14.79 | 44.02 | 69.64 | 70.01 | 54.21 | 64.35 | 3.67 | 16.31 | 69.56 |

Table 23: Ruler results on 32k context length.

| Exact | | Frozen | | | | | | Finetuned | | | |
|---|---|---|---|---|---|---|---|---|---|---|---|
| | Base ✓ | Inf | N-F ✓ | SE | PI ✓ | N-32 ✓ | YaRN ✓ | CLEX ✓ | LLR | Land | N-64 ✓ |
| Train Len | 4k | 4k | 4k | 4k | 32k | 32k | 32k | 32k | 32k | 32k | 64k |
| Eval Len | 4k | 4k | 4k | 4k | 4k | 4k | 4k | 4k | 4k | 4k | 4k |
| NIAH_S1 | - | 7.80 | 0.00 | 83.00 | 97.20 | 99.00 | 85.80 | 85.60 | 0.00 | 33.80 | 100.00 |
| NIAH_S2 | - | 7.00 | 0.00 | 68.40 | 99.00 | 100.00 | 81.00 | 94.00 | 0.00 | 3.20 | 99.20 |
| NIAH_S3 | - | 6.40 | 0.00 | 42.80 | 97.00 | 99.40 | 62.40 | 97.20 | 0.00 | 2.60 | 96.40 |
| NIAH_M1 | - | 8.80 | 0.00 | 29.40 | 93.40 | 90.80 | 63.20 | 78.40 | 0.00 | 5.40 | 82.60 |
| NIAH_M2 | - | 0.00 | 0.00 | 2.40 | 48.80 | 39.40 | 6.40 | 40.40 | 0.00 | 0.20 | 36.60 |
| NIAH_M3 | - | 0.00 | 0.00 | 1.40 | 5.80 | 8.60 | 1.20 | 8.00 | 0.00 | 0.00 | 7.20 |
| NIAH_MV | - | 3.75 | 0.00 | 24.65 | 61.30 | 68.20 | 37.95 | 60.75 | 0.00 | 2.80 | 82.20 |
| NIAH_MQ | - | 2.05 | 0.00 | 20.35 | 68.65 | 78.25 | 46.95 | 67.80 | 0.05 | 2.35 | 85.80 |
| VT | - | 2.08 | 0.00 | 2.32 | 56.68 | 43.28 | 22.00 | 30.08 | 0.00 | 2.52 | 71.28 |
| CWE | - | 4.48 | 0.02 | 17.46 | 26.72 | 11.78 | 11.38 | 22.70 | 13.26 | 3.68 | 7.34 |
| FWE | - | 72.67 | 2.93 | 46.73 | 31.00 | 64.53 | 15.13 | 34.67 | 13.93 | 72.47 | 51.53 |
| QA_1 | - | 20.20 | 5.20 | 20.60 | 33.40 | 34.40 | 23.00 | 28.00 | 6.00 | 22.60 | 27.00 |
| QA_2 | - | 25.20 | 1.20 | 24.00 | 30.60 | 34.80 | 24.00 | 30.60 | 12.60 | 24.60 | 33.20 |
| Avg. | - | 12.34 | 0.72 | 29.50 | 57.66 | 59.42 | 36.95 | 52.17 | 3.53 | 13.56 | 60.03 |

Table 24: Ruler results on 64k context length.

| Exact | Frozen | | | | Finetuned | | | | | | |
|---|---|---|---|---|---|---|---|---|---|---|---|
| | Base ✓ | Inf | N-F ✓ | SE | PI ✓ | N-32 ✓ | YaRN ✓ | CLEX ✓ | LLR | Land | N-64 ✓ |
| Train Len | 4k | 4k | 4k | 4k | 32k | 32k | 32k | 32k | 32k | 32k | 64k |
| Eval Len | 4k | 4k | 4k | 4k | 4k | 4k | 4k | 4k | 4k | 4k | 4k |
| NIAH_S1 | - | 3.20 | 0.00 | 71.80 | 0.00 | 83.60 | 0.00 | 40.60 | 0.00 | 40.00 | 98.00 |
| NIAH_S2 | - | 3.80 | 0.00 | 0.20 | 0.00 | 95.60 | 0.00 | 68.80 | 0.00 | 3.00 | 98.00 |
| NIAH_S3 | - | 5.40 | 0.00 | 0.00 | 0.00 | 95.40 | 0.00 | 70.40 | 0.00 | 3.00 | 95.80 |
| NIAH_M1 | - | 5.40 | 0.00 | 0.00 | 0.00 | 76.80 | 0.00 | 55.40 | 0.00 | 5.20 | 67.20 |
| NIAH_M2 | - | 0.00 | 0.00 | 2.60 | 0.00 | 15.20 | 0.00 | 15.80 | 0.00 | 0.00 | 25.80 |
| NIAH_M3 | - | 0.00 | 0.00 | 0.20 | 0.00 | 1.20 | 0.00 | 1.00 | 0.00 | 0.00 | 4.00 |
| NIAH_MV | - | 4.45 | 0.00 | 0.20 | 0.00 | 51.70 | 0.00 | 36.40 | 0.00 | 3.70 | 51.20 |
| NIAH_MQ | - | 4.45 | 0.00 | 0.05 | 0.00 | 56.60 | 0.00 | 43.50 | 0.00 | 2.45 | 65.40 |
| VT | - | 1.28 | 0.00 | 12.20 | 0.00 | 34.28 | 0.00 | 0.00 | 0.00 | 2.40 | 41.48 |
| CWE | - | 0.76 | 0.00 | 6.85 | 0.00 | 6.58 | 0.00 | 9.72 | 0.00 | 1.70 | 7.88 |
| FWE | - | 72.20 | 11.47 | 26.47 | 0.00 | 25.27 | 0.00 | 11.73 | 0.00 | 82.67 | 27.73 |
| QA_1 | - | 16.20 | 0.20 | 0.80 | 0.00 | 30.80 | 0.00 | 25.60 | 0.00 | 19.60 | 29.20 |
| QA_2 | - | 20.20 | 0.20 | 0.00 | 0.00 | 28.40 | 0.00 | 19.00 | 0.00 | 20.20 | 29.40 |
| Avg. | - | 10.56 | 0.91 | 9.34 | 0.00 | 46.26 | 0.00 | 30.61 | 0.00 | 14.15 | 49.31 |

