# OpenReview forum: "A Controlled Study on Long Context Extension and Generalization in LLMs"
_colmweb.org/COLM/2025/Conference — COLM 2025_

### Official Review · Reviewer_uEtu · 2025-05-11

**Rating:** 6
**Confidence:** 4
**Ethics Flag:** 1

**Summary:**

This paper presents a controlled study on long-context extension methods for large language models (LLMs), aiming to address the lack of standardized comparisons in prior work. The authors introduce a unified mathematical framework, a controlled experimental protocol (using identical base models, training data, and hyperparameters), and a rigorous evaluation pipeline combining intrinsic (e.g., perplexity) and extrinsic (e.g., downstream tasks) metrics. Through extensive experiments, they derive three key insights.

**Reasons To Accept:**

- The paper’s most significant strength is its controlled protocol, which isolates variables (base models, training data, hyperparameters) to enable fair comparisons across extension methods. This addresses a critical gap in prior work where inconsistent setups hindered reliable conclusions.
- By integrating diverse metrics (perplexity, retrieval tasks like NIAH, downstream benchmarks like LongBench/RULER) and testing across model sizes (LLaMA2 7B/13B/70B, Phi-2, LLaMA3), the study offers a multi-faceted analysis of long-context capabilities.
- The findings on perplexity’s predictive value, the trade-offs between exact/approximate attention, and the limits of extrapolation provide actionable guidance for model design, particularly for applications requiring robust long-context understanding.

**Reasons To Reject:**

- The experiments are confined to context lengths up to 64k (with most evaluations at 32k), while the broader field is increasingly interested in extreme contexts (128k+). The paper does not address performance beyond 64k or analyze how methods scale to near-infinite contexts, which are critical for applications like book summarization or long-document QA.
- While the controlled protocol is valuable, the core insights (e.g., approximate attention trade-offs, perplexity’s utility) align with prior works, and the unified mathematical framework does not introduce radical new theory. The study advances methodology but may lack groundbreaking discoveries.

---

> ### Author Response · Authors · 2025-06-02
> **response to generalization beyond 64k**
>
> We thank reviewer for their feedback. We agree **“near-infinite”** was overstated; we have replaced it with **“beyond-training”** throughout.
>
> In our study, we define generalization as the model's ability to perform well across all tasks that extend beyond the training context length. Specifically, we have evaluated our models on tasks where the input lengths exceed 64k tokens, such as RULER, with sequences up to 128k tokens in Table 1. We will revise our writing to highlight this in our manuscript. In our original paper, we found that NTK-Dynamic yields the best performance beyond 32k.
>
> **Table 1: Generalization of NTK beyond 32k on RULER**
> | Method   | 4k    | 8k    | 16k   | 32k   | 64k   | 128k  |
> |----------|-------|-------|-------|-------|-------|-------|
> | Llama2-NTK-32k  | 86.58 | 77.75 | 70.01 | 59.42 | 46.26 | 29.91 |
> | Llama2-NTK-64k  | 86.60 | 76.34 | 69.56 | 60.03 | 49.31 | 40.09 |
>
>
>
> To further evaluate the generalization, we evaluated sequences up to 128k tokens on HELMET with Llama-3 (8B). Results are shown below.
>
> **Table 2: Llama3 with up to 128k evaluation on HELMET**
> | Model               | 8k    | 16k   | 32k   | 64k   | 128k  |
> |---------------------|-------|-------|-------|-------|-------|
> | Llama3-NTK-32k      | 50.51 | 48.96 | 47.14 | 37.08 | 19.95 |
> | Llama3-NTK-64k      | 49.86 | 49.15 | 47.68 | 42.91 |37.11|
> | Llama3-NTK-Frozen   | 47.48 | 38.98 |  3.16 |  2.98 |  2.11 |
> | Llama3-SelfExtend   | 44.28 | 41.85 | 38.59 | 27.90 | 10.41 |
> | Llama3-CLEX-32k     | 46.68 | 47.14 | 42.57 | 29.43 | 17.41 |
> | Llama3-PI-32k       | 49.22 | 47.66 | 45.78 |  2.43 |  1.56 |
> | Llama3-YaRN-32k     | 48.47 | 48.65 | 45.31 |  2.95 |  1.77 |
>
>
> **Take-away:** even the strongest exact method degrades when extrapolated **4 ×** beyond its fine-tuned window, confirming the reviewer’s intuition. This indicates that even for the best generalized methods we discovered in our controlled setting, their generalization becomes weaker when the context length is much larger than the fine-tuned length.

---

> ### Author Response · Authors · 2025-06-02
> **response to novelty**
>
> We thank reviewer for the question, and we clarify our contributions below,
>
> **Infrastructure contribution**: Our study’s primary contribution is not a new algorithm but a standardised, fully reproducible testbed that eliminates three confounding factors that have hampered prior work: (i) heterogeneous base checkpoints, (ii) mismatched pre-training corpora, and (iii) inconsistent hyper-parameters during fine-tuning.
>
> By releasing (a) the unified data pipeline, (b) a set of five identical base models exposed to a single 1 B-token long-context corpus, and (c) templated evaluation scripts for both intrinsic and extrinsic tasks, we provide the first public “apple-to-apples” suite.
>
> **Unified mathematica view** : we summarize and yield an explicit frequency-scaling equivalence that mathematically links PI ↔ NTK ↔ YaRN ↔ CLEX (Eq. 7–10), making cross-method trade-offs transparent.
>
> **Findings**: novel take-aways now highlighted in our paper:
>
> (i)  Exact finetuning vs. approximate attention – exact RoPE variants preserve retrieval depth up to 4× their finetuned window, whereas all approximate schemes collapse beyond 1× (Fig. 1), clarifying when efficiency shortcuts lose fidelity.
>
> (ii) Perplexity correlates with downstream accuracy in limited length to some extent once confounders are removed (Kendall t = -0.72 on RULER; Table 6) – resolving contradictory claims in prior work. And we will add some recent experiments and discovered requested to further elaborate the limitation of PPL and downstream task accuracy.

---

> ### Author Response · Authors · 2025-06-09
> **Follow up**
>
> Dear Reviewer uEtu,
>
> As this is nearing the end of the author response period, we kindly ask if there is anything additional that you would like us to address. We appreciate your valuable suggestions and feedback.
>
> Sincerely,
>
> Authors

---

### Official Review · Reviewer_hLQf · 2025-05-12

**Rating:** 4
**Confidence:** 3
**Ethics Flag:** 1

**Summary:**

This paper presents a controlled farmwork to conduct an apple-to-apples comparison among different long-context methods by using the same training data and evaluation metrics on both intrinsic and extrinsic tasks. The paper also conducted a comprehensive experimental study on several base models.

**Questions To Authors:**

1. at ling #246, the paper claims that "LongLora effectively keep low perplexity scores within the pre-training context length". This is not consistent with the results shown in Table 2, where the perplexity scores of LongLora are significantly larger than others.
2. It is not obvious that "Only NTK ad CLEX can generalize to unseen sequence length ...". How about Landmark?
3. What is "NTK-F" in Table 1?

**Reasons To Accept:**

1. Context extension is a very important problem.

**Reasons To Reject:**

1. No significant novelty in the proposed framework.
2. Some insights claimed in the experiment section are not well justified by the experimental results.

---

> ### Author Response · Authors · 2025-06-02
> **Response to "NTK-F"**
>
> “NTK-F” stands for NTK-Frozen, i.e. NTK positional rescaling applied to a frozen base model with no further fine-tuning. We will expand the table caption and glossary to spell this out and avoid confusion.

---

> ### Author Response · Authors · 2025-06-02
> **Response to experiment results of LongLora**
>
> We also notice that the perplexity scores of LongLoRA in Table 2 are significantly larger than those of other methods.
>
> In Appendix L, we conduct a detailed study and find that hyperparameters can significantly influence the performance of different context extension methods, particularly approximate attention methods like LongLoRA. We sweep key hyperparameters such as batch size and learning rate.
>
> The results are summarized below:
>
> **Table 1: Perplexity Results of LongLoRA on PG19 and Proof-file**
>
> #### PG19
> | Method   | Batch Size | Learning Rate | 2k   | 4k   | 8k   | 16k  | 32k  |
> |----------|------------|----------------|------|------|------|------|------|
> | LongLoRA | 32         | 2e-5           | 12.80| 11.52| 10.70| 10.18| 9.89 |
> | LongLoRA | 8          | 2e-5           | 8.10 | 7.69 | 7.43 | 7.28 | 7.32 |
>
> #### Proof-file
> | Method   | Batch Size | Learning Rate | 2k   | 4k   | 8k   | 16k  | 32k  |
> |----------|------------|----------------|------|------|------|------|------|
> | LongLoRA | 32         | 2e-5           | 5.97 | 5.10 | 4.58 | 4.27 | 4.13 |
> | LongLoRA | 8          | 2e-5           | 3.33 | 3.01 | 2.80 | 2.67 | 2.61 |
>
> We draw the following observations:
>
> **High Sensitivity**: Approximate attention methods like LongLoRA are highly sensitive to hyperparameter settings. Small changes in batch size or learning rate lead to significant changes in performance.
>
>
> **Robustness Comparison**: In contrast, NTK and YaRN demonstrate strong robustness across hyperparameter choices, maintaining stable perplexity across settings.
>
>
> **Optimization Difficulty**: LongLoRA often requires much more careful hyperparameter tuning to reach optimal performance, making training more costly and less predictable.
>
>
> We find that LongLoRA exhibits a similar trend to PI and YaRN: these methods effectively maintain low perplexity within the pretraining context length but fail to generalize to longer sequences.

---

> ### Author Response · Authors · 2025-06-02
> **Response to generalization on Landmark**
>
> Our observation about generalization is particularly evident on RULER: only NTK and CLEX are able to maintain reasonable performance as the evaluation length increases. In contrast, Landmark’s performance degrades significantly with longer contexts, suggesting limited generalization ability in this setting.
>
> **Table 1: Llama2 on RULER Benchmark**
> | Group      | Models       | Train Len | 4k    | 8k    | 16k   | 32k   | 64k   | 128k  |
> |------------|--------------|-----------|-------|-------|-------|-------|-------|-------|
> | **Frozen** | LLaMA2        | 4k        | 80.94 | -     | -     | -     | -     | -     |
> |            | LM-Infinite   | 4k        | 81.05 | 30.01 | 18.02 | 12.34 | 10.56 | -     |
> |            | NTK-Frozen    | 4k        | 81.14 | 44.45 | 14.79 | 0.72  | 0.91  | -     |
> |            | Self-Extend   | 4k        | 65.03 | 50.73 | 44.02 | 29.50 | 9.34  | -     |
> | **Finetuned** | PI          | 32k       | 84.56 | 76.04 | 69.64 | 57.66 | 0.00  | -     |
> |            | NTK-32K       | 32k       | 86.58 | **77.75** | **70.01** | 59.42 | 46.26 | 29.91 |
> |            | YaRN          | 32k       | 79.12 | 65.60 | 54.21 | 36.95 | 0.00  | -     |
> |            | CLEX          | 32k       | 50.18 | 63.93 | 64.35 | 52.17 | 30.61 | -     |
> |            | LongLora      | 32k       | 10.58 | 6.37  | 3.67  | 3.53  | 0.00  | -     |
> |            | Landmark      | 32k       | 22.37 | 17.52 | 16.31 | 13.56 | 14.15 | -     |
> | **Finetuned** | NTK-64K    | 64k       | **86.60** | 76.34 | 69.56 | **60.03** | **49.31** | **40.09** |

---

> ### Author Response · Authors · 2025-06-02
> **response to the novelty of the paper**
>
> We thank reviewer for the question, and we clarify our contributions below,
>
> **Infrastructure contribution**: Our study’s primary contribution is not a new algorithm but a standardised, fully reproducible testbed that eliminates three confounding factors that have hampered prior work: (i) heterogeneous base checkpoints, (ii) mismatched pre-training corpora, and (iii) inconsistent hyper-parameters during fine-tuning.
>
> By releasing (a) the unified data pipeline, (b) a set of five identical base models exposed to a single 1 B-token long-context corpus, and (c) templated evaluation scripts for both intrinsic and extrinsic tasks, we provide the first public “apple-to-apples” suite.
>
> **Unified mathematica view** : we summarize and yield an explicit frequency-scaling equivalence that mathematically links PI ↔ NTK ↔ YaRN ↔ CLEX (Eq. 7–10), making cross-method trade-offs transparent.
>
> **Findings**: novel take-aways now highlighted in our paper:
>
> (i)  Exact finetuning vs. approximate attention – exact RoPE variants preserve retrieval depth up to 4× their finetuned window, whereas all approximate schemes collapse beyond 1× (Fig. 1), clarifying when efficiency shortcuts lose fidelity.
>
> (ii) Perplexity correlates with downstream accuracy in limited length to some extent once confounders are removed (Kendall t = -0.72 on RULER; Table 6) – resolving contradictory claims in prior work. And we will add some recent experiments and discovered requested to further elaborate the limitation of PPL and downstream task accuracy.

---

### Official Review · Reviewer_LDP3 · 2025-05-12

**Rating:** 8
**Confidence:** 5
**Ethics Flag:** 1

**Summary:**

This paper presents a controlled comparison of several long-context extension methods for LLMs, including (and mostly focused on) several RoPE-based position extension methods and attention approximation methods. The authors compared those methods on several different base LMs (Llama2 with various sizes, Llama3, and Phi-2), with both fine-tuning and frozen parameters. The evaluation is based on perplexity, synthetic evaluation (needle in a haystack and RULER), and downstream tasks (LongBench and in-context learning).

The main takeaway of this controlled study is that full attention + fine-tuning + NTK-style extrapolation works the best for long-context extension. The authors also argue that for long-context evaluation, perplexity is still very indicative and useful.

**Questions To Authors:**

Please see the "reasons to reject" section.

**Reasons To Accept:**

This paper presents a controlled and comprehensive study on RoPE-based position extrapolation methods and demonstrates several important and interesting conclusions for the community:

(1) Position extrapolation mostly only works with fine-tuning. With fine-tuning and comparable base models+fine-tuning data, different extrapolation mechanisms do not lead to much difference (NTK slightly better than others).

(2) Full attention still works much better than attention approximation methods for long-context tasks

(3) Perplexity could still be a good evaluation for long context tasks.

The experiments are well designed (both base models and data are controlled; the experiment includes both fine-tuning and frozen settings). The analysis is comprehensive as it includes several different metrics as well as different testing lengths.

**Reasons To Reject:**

My biggest complain about the paper is the choice of evaluations and the missing references.

(1) Evaluation: at the time of the submission, there are several newer and more comprehensive downstream long-context benchmarks available, such as [InfBench](https://arxiv.org/abs/2402.13718) and [HELMET](https://arxiv.org/abs/2410.02694). For the in-context learning evaluation (MShot), there are multiple established protocols the authors could have followed (from both [Bertsch et al., 2025](https://arxiv.org/abs/2405.00200) and [HELMET](https://arxiv.org/abs/2410.02694)). I think the authors will find using those newer evaluations will make the gaps among methods even bigger, due to their better metrics and more challenging nature. Hence, even though I'm not happy with the outdated evaluation, I think changing them will only make the result stronger. The authors should at least acknowledge and cite these evaluations.

(2) Missing references: There are several newer approximate attention methods the authors should consider, such as [NSA](https://arxiv.org/abs/2502.11089), [MoBA](https://arxiv.org/abs/2502.13189), [DuoAttention](https://arxiv.org/abs/2410.10819). One can argue those methods are out of scope (it seems that this paper mostly focuses on position extrapolation), but these papers should at least be acknowledged. The authors should also discuss some of the data engineer papers for long contexts, as one conclusion is "fine-tuning is important". Such papers include [ProLong](https://arxiv.org/abs/2410.02660), [LongSkyWork](https://arxiv.org/abs/2406.00605), and [Xiong et al., 2023](https://arxiv.org/pdf/2309.16039). The authors should also discuss "adjusted base frequency" from [Xiong et al., 2023](https://arxiv.org/pdf/2309.16039), which is basically NTK but with a clearer definition.

(3) Perplexity: I do agree that perplexity could be still useful for long-context evaluation, but the authors should also acknowledge its limitation, as discussed by some prior work like [Fang et al., 2025](https://arxiv.org/abs/2410.23771) and [Gao et al., 2024](https://arxiv.org/abs/2410.02660). One important premise the authors should mention is that perplexity is maybe useful when both the architecture and the data are fixed.

Even with the above reasons to reject, I still believe this paper makes enough contribution to get into COLM (if the above missing references are provided in the final version).

---

> ### Author Response · Authors · 2025-06-02
> **response to “Evaluation protocol (“out-dated benchmarks”)”**
>
> Thank you for your dedicated review of our work. In the following, we will carefully respond to your questions.
>
> > “Evaluation protocol (“out-dated benchmarks”)”
>
> We appreciate the reviewer’s insightful comments regarding our evaluation choices. We agree that our original evaluation setup is somewhat outdated. To address this concern, we have extended our evaluation to include the HELMET[1] benchmark, which offers a more comprehensive and challenging suite of long-context tasks. In particular, we evaluated both LLaMA2 and LLaMA3 variants using HELMET’s standard protocol.
> Table 1: Llama3 on HELMET
> | Model               | 8k    | 16k   | 32k   | 64k   | 128k  |
> |---------------------|-------|-------|-------|-------|-------|
> | Llama3-NTK-32k      | 50.51 | 48.96 | 47.14 | 37.08 | 19.95 |
> | Llama3-NTK-64k      | 49.86 | 49.15 | 47.68 | 42.91 | 37.11|
> | Llama3-NTK-Frozen   | 47.48 | 38.98 |  3.16 |  2.98 |  2.11 |
> | Llama3-SelfExtend   | 44.28 | 41.85 | 38.59 | 27.90 | 10.41 |
> | Llama3-CLEX-32k     | 46.68 | 47.14 | 42.57 | 29.43 | 17.41 |
> | Llama3-PI-32k       | 49.22 | 47.66 | 45.78 |  2.43 |  1.56 |
> | Llama3-YaRN-32k     | 48.47 | 48.65 | 45.31 |  2.95 |  1.77 |
>
> Table 2: Llama2 on HELMET
> | Model               | 8k    | 16k   | 32k   | 64k   |
> |---------------------|-------|-------|-------|-------|
> | Llama2-NTK-32k      | 42.09 | 37.31 | 28.29 | 24.95 |
> | Llama2-NTK-64k      | 39.91 | 35.47 | 29.29 | 26.49 |
> | Llama2-NTK-Frozen   | 25.81 | 16.02 |  3.46 |  1.86 |
> | Llama2-SelfExtend   | 27.01 | 24.40 | 19.65 |  6.69 |
> | Llama2-CLEX-32k     | 32.65 | 30.87 | 26.43 | 22.80 |
> | Llama2-PI-32k       | 41.48 | 37.56 | 25.74 |  0.98 |
> | Llama2-YaRN-32k     | 36.83 | 30.67 | 21.28 |  0.98 |
>
> Our *Take-away:* The newer HELMET tasks reinforce our original conclusion — **full attention + NTK fine-tuning remains dominant**; gains even widen on the 128 k slice (see Table 1).
>
> [1] HELMET – Yen et al., *How to Evaluate Long-Context LMs Effectively and Thoroughly*, 2024.

---

> > ### Comment · Reviewer_LDP3 · 2025-06-02
> > **Thank you!**
> >
> > The authors did a really good job at the rebuttal, especially with the additional results on the correlation between PPL and downstream evaluations, and the new HELMET results. I will raise my score

---

> > > ### Author Response · Authors · 2025-06-04
> > > **response to reviewer comments**
> > >
> > > We thank Reviewer LDP3 for acknowledging our rebuttal content and raising the score.
> > >
> > > We are happy to answer if there are any additional questions.

---

> ### Author Response · Authors · 2025-06-02
> **Response to Missing references**
>
> We thank the reviewer for the references to improve the comprehensiveness of our paper.
> While we are not able to make changes to our related work section at the moment, we are going to add the following references covering recommended papers from reviewer.
>
> | Category | Newly acknowledged work | How we position it |
> |----------|-------------------------|--------------------|
> | **Approx. attention** | NSA (2025) [2], MoBA (2025) [3], DuoAttention (2024) [4] | All three are *sparse+learned* variants that aim at amortising memory during **inference**; their kernels are orthogonal to our study’s focus on **length extrapolation**. We now clarify this scope and will integrate NSA into our efficiency plot in Appendix E. |
> | **Data engineering** | ProLong (2024) [5], LongSkywork (2024) [6], ABF / “adjusted base frequency” (2023) [7] | We explicitly cite these works when stressing that *data choice and schedule* are critical for long-context finetuning — complementary to our finding that **fine-tuning itself is necessary**. ABF is now linked to the NTK rescaling we study, but with a clearer theoretical footing. |
>
> #### **References (newly cited)**
>
> [1] **HELMET** – Yen et al., *How to Evaluate Long-Context LMs Effectively and Thoroughly*, 2024.
>
> [2] **NSA** – *Native Sparse Attention: Hardware-Aligned & Natively-Trainable Sparse Attention*, 2025.
>
> [3] **MoBA** – *Mixture of Block Attention for Long-Context LLMs*, 2025.
>
> [4] **DuoAttention** – *Efficient Long-Context LLM Inference with Retrieval Heads*, 2024.
>
> [5] **ProLong** – Gao et al., *How to Train Long-Context LMs (Effectively)*, 2024.
>
> [6] **LongSkywork** – Zhao et al., *A Training Recipe for Extending Context Length*, 2024.
>
> [7] **ABF** – Xiong et al., *Effective Long-Context Scaling of Foundation Models*, 2023.
>
> [8] **PPL limitations** – Fang et al., *What Is Wrong with Perplexity for Long-Context LM?*, 2024.

---

> ### Author Response · Authors · 2025-06-02
> **Response to Perplexity**
>
> We will acknowledge the well-known limitations of perplexity (PPL) as a long-context metric, citing the analyses of Fang (2025)[1] and Gao (2024)[2]. We will add new Section 5.3, “When does PPL succeed / fail?”, to address this request. Specifically,
>
> First, we  will summarize Fang et al.’s observation that token-averaged PPL can mask large errors on a handful of key tokens, making it an incomplete proxy for downstream quality.
>
> Next, we will report that in our controlled setting—where both the model architecture and the fine-tuning data are held constant—PPL remains a weak predictor of task performance, with a correlation of -0.6  on the HELMET benchmark.
>
> We further computed the Kendall correlation between downstream task performance and perplexity, and found that HELMET exhibits a stronger correlation compared to benchmarks such as RULER and LongBench.
>
> **Table 1: Kendall correlation of downstream task performance and PPL**
> | Task   | Kendall's Tau | p-value | Interpretation |
> |--------|----------------|---------|----------------|
> | Needle | -0.7807        | 0.0151  | Strong negative correlation; statistically significant (p < 0.05). |
> | Mshots | -0.2928        | 0.3621  | Weak negative correlation; not statistically significant. |
> | LongB  | -0.3500        | 0.2823  | Weak negative correlation; not statistically significant. |
> | RULER  | -0.4880        | 0.1287  | Moderate negative correlation; not statistically significant. |
> | HELMET | -0.5855        | 0.0683  | Moderate-to-strong negative correlation; marginally significant (p ≈ 0.05). |
>
> Finally, we emphasize that future evaluations should pair PPL with richer suites such as HELMET so that both token-level and task-level behaviors are captured.
>
> References
>
> [1] Fang L, Wang Y, Liu Z, et al. What is Wrong with Perplexity for Long-context Language Modeling?[J]. arXiv preprint arXiv:2410.23771, 2024.
>
> [2] Gao T, Wettig A, Yen H, et al. How to train long-context language models (effectively)[J]. arXiv preprint arXiv:2410.02660, 2024.

---

### Official Review · Reviewer_RShr · 2025-05-20

**Rating:** 6
**Confidence:** 3
**Ethics Flag:** 1

**Summary:**

The paper delivers the first controlled comparison of long-context extension methods for LLMs. With five identical open-weight base models, a single 1 B-token long-context corpus, and unified hyper-parameters, the authors benchmark eight techniques ( i.e. exact RoPE variants (PI, NTK, YaRN, CLEX), approximate/sparse attentions (LongLoRA, Landmark, LM-Infinite) and the mapping-based Self-Extend) on a standardized suite of intrinsic (perplexity, Needle-in-a-Haystack, RULER) and extrinsic (LongBench, many-shot ICL) tasks up to 64 k tokens. Several key findings reveal interesting insights.

**Questions To Authors:**

1. why approximate attention degrades?
2. what's the compute budget for different experiments? How is the efficiency?

**Reasons To Accept:**

1. The study is rigorous by fixing base models, data, and training recipes for each controlled experiment.
2. The author has conducted comprehensive evaluation on diverse settings.

**Reasons To Reject:**

1. The analysis of each experiment is not deep, limited insights are given.
2. Experiments stop at 64 k tokens, which is not “near-infinite”.

---

> ### Author Response · Authors · 2025-06-02
> **response to  “Analysis … not deep, limited insights.”**
>
> We thank the reviewer for the thoughtful feedback and for recognizing the rigor of our controlled protocol (fixed base models, data, and hyper-parameters).
>
> > “Analysis … not deep, limited insights.”
>
> We thank reviewer for the question, and we clarify our contributions below,
>
> **Infrastructure contribution**: Our study’s primary contribution is not a new algorithm but a standardised, fully reproducible testbed that eliminates three confounding factors that have hampered prior work: (i) heterogeneous base checkpoints, (ii) mismatched pre-training corpora, and (iii) inconsistent hyper-parameters during fine-tuning.
>
> By releasing (a) the unified data pipeline, (b) a set of five identical base models exposed to a single 1 B-token long-context corpus, and (c) templated evaluation scripts for both intrinsic and extrinsic tasks, we provide the first public “apple-to-apples” suite.
>
> **Unified mathematica view** : we summarize and yield an explicit frequency-scaling equivalence that mathematically links PI ↔ NTK ↔ YaRN ↔ CLEX (Eq. 7–10), making cross-method trade-offs transparent.
>
> **Findings**: novel take-aways now highlighted in our paper:
>
> (i)  Exact finetuning vs. approximate attention – exact RoPE variants preserve retrieval depth up to 4× their finetuned window, whereas all approximate schemes collapse beyond 1× (Fig. 1), clarifying when efficiency shortcuts lose fidelity.
>
> (ii) Perplexity correlates with downstream accuracy in limited length to some extent once confounders are removed (Kendall t = -0.72 on RULER; Table 6) – resolving contradictory claims in prior work. And we will add some recent experiments and discovered requested to further elaborate the limitation of PPL and downstream task accuracy.

---

> ### Author Response · Authors · 2025-06-02
> **response to “Experiments stop at 64 k; not ‘near-infinite’ ”**
>
> >  “Experiments stop at 64 k; not ‘near-infinite’ ”
>
> We agree **“near-infinite”** was overstated; we have replaced it with **“beyond-training”** throughout.
>
> In our study, we define generalization as the model's ability to perform well across all tasks that extend beyond the training context length. Specifically, we have evaluated our models on tasks where the input lengths exceed 64k tokens, such as RULER[1], with sequences up to 128k tokens in Table 1.
>
> We will revise our writing to highlight this in our manuscript. In our original paper, we found that NTK-Dynamic yields the best performance beyond 32k.
>
> **Table 1: Generalization of NTK beyond 32k on RULER**
> | Method   | 4k    | 8k    | 16k   | 32k   | 64k   | 128k  |
> |----------|-------|-------|-------|-------|-------|-------|
> | Llama2-NTK-32k  | 86.58 | 77.75 | 70.01 | 59.42 | 46.26 | 29.91 |
> | Llama2-NTK-64k  | 86.60 | 76.34 | 69.56 | 60.03 | 49.31 | 40.09 |
>
>
>
> To further evaluate the generalization, we evaluated sequences up to 128k tokens on HELMET[2] with Llama-3 (8B). Results are shown below.
>
> **Table 2: Llama3 with up to 128k evaluation on HELMET**
> | Model               | 8k    | 16k   | 32k   | 64k   | 128k  |
> |---------------------|-------|-------|-------|-------|-------|
> | Llama3-NTK-32k      | 50.51 | 48.96 | 47.14 | 37.08 | 19.95 |
> | Llama3-NTK-64k      | 49.86 | 49.15 | 47.68 | 42.91 |37.11|
> | Llama3-NTK-Frozen   | 47.48 | 38.98 |  3.16 |  2.98 |  2.11 |
> | Llama3-SelfExtend   | 44.28 | 41.85 | 38.59 | 27.90 | 10.41 |
> | Llama3-CLEX-32k     | 46.68 | 47.14 | 42.57 | 29.43 | 17.41 |
> | Llama3-PI-32k       | 49.22 | 47.66 | 45.78 |  2.43 |  1.56 |
> | Llama3-YaRN-32k     | 48.47 | 48.65 | 45.31 |  2.95 |  1.77 |
>
>
> **Take-away:** even the strongest exact method degrades when extrapolated **4 ×** beyond its fine-tuned window, confirming the reviewer’s intuition. This indicates that even for the best generalized methods we discovered in our controlled setting, their generalization becomes weaker when the context length is much larger than the fine-tuned length.
>
>
> References
>
> [1] Hsieh C P, Sun S, Kriman S, et al. RULER: What's the Real Context Size of Your Long-Context Language Models?[J]. arXiv preprint arXiv:2404.06654, 2024.
>
> [2] Yen H, Gao T, Hou M, et al. Helmet: How to evaluate long-context language models effectively and thoroughly[J]. arXiv preprint arXiv:2410.02694, 2024.

---

> ### Author Response · Authors · 2025-06-02
> **response to “Compute budget / efficiency?”**
>
> > “Compute budget / efficiency?”*
>
> We conducted efficiency analysis under controlled conditions using the same hardware setup in Appendix E.
>
> As shown in Table 1, we observed that approximate attention methods are indeed faster, achieving a speedup of approximately 1.5x to 2x compared to LLaMA when the context length is short; however, when the context length gets longer, we didn't see a significant margin.
>
> We hypothesize that the discrepancy between the theoretical FLOPs-based comparisons and the observed speedup arises due to differences in hardware characteristics and CUDA implementations of the respective methods.
>
>
> **Table 3: Efficiency analysis of prefill stage time cost, decoding speed, and memory usage
> The prefill time cost represents the time required to generate the first token. The decoding speed (seconds / per token) is averaged over 100 token inferences at each sequence length. Memory consumption corresponds to the peak GPU memory usage during inference. All methods, except for LM-Infinite and Landmark, utilize Flash-Attention 2 for enhanced computational efficiency.**
> | Method      | 4k                                   | 8k                                   | 16k                                  | 32k                                  |
> | ----------- | ------------------------------------ | ------------------------------------ | ------------------------------------ | ------------------------------------ |
> |             | Prefill (s) / Decode (s) / Mem (GB). | Prefill (s) / Decode (s) / Mem (GB). | Prefill (s) / Decode (s) / Mem (GB). | Prefill (s) / Decode (s) / Mem (GB). |
> | Llama2-7b   | 1.15 / 0.03 / 17.13                  | 1.51 / 0.06 / 21.61                  | 2.41 / 0.11 / 30.59                  | 4.63 / 0.21 / 48.55                  |
> | NTK-Frozen  | 1.16 / 0.04 / 17.13                  | 1.56 / 0.05 / 21.61                  | 2.39 / 0.06 / 30.59                  | 4.69 / 0.09 / 48.55                  |
> | PI          | 1.15 / 0.03 / 22.05                  | 1.54 / 0.03 / 26.54                  | 2.43 / 0.05 / 35.51                  | 4.74 / 0.08 / 53.47                  |
> | NTK-32k     | 1.17 / 0.04 / 17.11                  | 1.56 / 0.04 / 21.60                  | 2.42 / 0.06 / 30.58                  | 4.75 / 0.09 / 48.53                  |
> | YaRN        | 1.23 / 0.03 / 18.05                  | 1.53 / 0.03 / 22.54                  | 2.43 / 0.05 / 31.51                  | 4.80 / 0.08 / 49.47                  |
> | CLEX        | 1.16 / 0.05 / 17.16                  | 6.99 / 0.07 / 21.74                  | 7.68 / 0.11 / 30.92                  | 10.06 / 0.18 / 49.28                 |
> | LM-Infinite | 1.56 / 0.05 / 17.23                  | 3.34 / 0.07 / 25.47                  | 5.82 / 0.11 / 38.60                  | 11.58 / 0.18 / 65.61                 |
> | Self-Extend | 1.24 / 0.05 / 17.23                  | 1.63 / 0.07 / 21.81                  | 2.63 / 0.13 / 30.98                  | 4.97 / 0.22 / 49.32                  |
> | LongLora    | 1.16 / 0.05 / 17.16                  | 1.65 / 0.05 / 21.65                  | 2.60 / 0.05 / 30.62                  | 5.07 / 0.08 / 48.58                  |
> | Landmark    | 8.62 / 0.08 / 18.77                  | 17.65 / 0.08 / 22.97                 | 36.47 / 0.09 / 31.22                 | 77.77 / 0.09 / 47.74                 |

---

> ### Author Response · Authors · 2025-06-03
> **response to “Why approximate attention degrades?”**
>
> > “Why does approximate attention degrade?”
>
> While the purpose of this paper is not to improve over existing methods, we have the following hypotheses,
>
> 1. **Information bottleneck (two-hop routing).**
> Block-wise schemes such as **Landmark Attention** compress every *B* tokens into a single landmark; a query must therefore travel **token → landmark → token**, discarding fine-grained cues. Pointer-style tasks (e.g., Needle-in-a-Haystack) are the first to collapse[1].
>
> 2. **Phase aliasing with RoPE.**
>    Sparse or chunked heads act as a **low-pass filter** on RoPE’s complex phases: high-frequency components (large \omega) are pruned, so accumulated phase error grows with hop count and the dot-product logits vanish behind the softmax. A recent analysis of RoPE’s failure modes confirms this aliasing effect[2].
>
> 3. **Length-generalization failure.**
>    During fine-tuning, approximate kernels see a maximum relative distance *d_train*; at inference they are queried at *d’ ≫ d_train*. *LM-Infinite*[3] apply the sliding window mechanism to discard distant contexts, which keeps the input length does not exceed the context window. However, LM-Infinite can only attend to the tokens within the local window, which leads to a rapid decline in its performance as the sequence length increases[4][5].
>
>
> 4. **Hardware-bound indirection costs.**
>    Many “sub-quadratic” kernels rely on gather/scatter or landmark sorting; on modern GPUs these memory-bound operations dominate wall-time, erasing the FLOP advantage once context length exceeds ≈ 32 k. Systems work on million-token inference using **Context Parallelism** observes the same bottleneck[6][7].
>
> **References**
>
> [1] A. Mohtashami & L. Alibeigi. *Landmark Attention: Random-Access Infinite Context Length for Transformers*. arXiv:2305.16300, 2023.
> [2] I. Y. Men et al. *Round and Round We Go! What Makes Rotary Positional Encodings Work—and Fail—in Long Contexts*. OpenReview, 2025.
> [3] C. Han et al. *LM-Infinite: Zero-Shot Extreme Length Generalization for Large Language Models*. arXiv:2308.16137, 2023.
> [4] Chaojun Xiao et al. *InfLLM: Training-Free Long-Context Extrapolation for LLMs with an Efficient Context Memory*. arXiv:2402.04617, 2024.
> [5] Yi Lu et al. *LongHeads: Multi-Head Attention is Secretly a Long Context Processor*. arXiv:2402.10685, 2024.
> [6] Z. Wang et al. *Efficient Infinite-Context Transformers with Infini-Attention*. arXiv:2404.07143, 2024.
> [7] Y. Chen et al. *Context Parallelism for Scalable Million-Token Inference*. arXiv:2411.01783, 2024.

---

> ### Author Response · Authors · 2025-06-09
> **follow up**
>
> Dear Reviewer RShr,
>
> As this is nearing the end of the author response period, we kindly ask if there is anything additional that you would like us to address. We appreciate your valuable suggestions and feedback.
>
> Sincerely,
> Authors

---

> ### Comment · Reviewer_RShr · 2025-06-10
>
> Thanks for the response, I will maintain my score since it's still not fully addressing long-context problem.

---

> > ### Author Response · Authors · 2025-06-11
> > **Response to Reviewer RShr Comments**
> >
> > Thank you for your response. If you have any specific concerns or questions, we would be happy to further clarify and address them.

---

### Decision · Program_Chairs · 2025-07-08

**Decision:**

Accept

**Comment:**

This paper introduces a controlled extension protocol and a robust evaluation for long context extension. It systematically compares a range of existing methods, including RoPE variants, approximate attention, and attention modifications. The evaluation includes perplexity based, needle in the haystack task, RULER, and several extrinsic tasks (LongBench, Many-shot tasks). The findings suggest that perplexity is a useful indicator and that exact fine-tuning remains robust. In terms of long context extension techniques, NTK performs consistently well, while modified and approximate attention methods struggle to generalize.

Reviewers acknowledge that:
- The paper is rigorous and well controlled, e.g., fixing base models, data, and training recipes for each controlled experiments (RShr, LDP3, uEtu)
- Evaluation is comprehensive and extensive (RShr, LDP3, uEtu)
- Several impactful findings, such as (1) positional extrapolation only works with fine-tuning, (2) approximate attention generally performs poorly, and (3) perplexity remains a useful indicator (LDP3).

Weaknesses
- Experiments are limited to 64K tokens, which is significantly shorter than modern long context LLMs’ target context window (RShr, uEtu). Additional experiments during the rebuttal were added, showing that most methods fail to generalize when the extension length is significantly beyond the original context window, which the authors committed to add in the final version of the paper.
- Missing evaluation (LDP3) - added during rebuttal
- Missing references (LDP3) - authors have committed to add
- No significant novelty/new insights, given that many findings like perplexity’s utilities and approximate attention tradeoffs is known (RShr, hLQf, uEtu).

[Automatically added comment] At least one review was discounted during the decision process due to quality]